# Variability and drivers of winter near-surface temperatures over boreal and tundra landscapes

Vilna Tyystjärvi[1,2], Pekka Niittynen[3], Julia Kemppinen[4], Miska Luoto[2], Tuuli Rissanen[5], and Juha Aalto[6,2]

[1]Climate System Research, Finnish Meteorological Institute, P.O. Box 503, 00101 Helsinki, Finland
[2]Department of Geosciences and Geography, University of Helsinki, P.O. Box 64, 00014 University of Helsinki, Finland
[3]Department of Biological and Environmental Science, University of Jyväskylä, PL 35, 40014 University of Jyväskylä, Finland
[4]The Geography Research Unit, University of Oulu, P.O. Box 8000, 90014 University of Oulu, Finland
[5]Research Centre for Ecological Change, Organismal and Evolutionary Biology Research Programme, University of Helsinki, P.O. Box 64, 00014 University of Helsinki, Finland
[6]Weather and Climate Change Impact Research, Finnish Meteorological Institute, P.O. Box 503, 00101 Helsinki, Finland
**Correspondence:** Vilna Tyystjärvi (vilna.tyystjarvi@helsinki.fi)

**Abstract.** Winter near-surface air temperatures have important implications for ecosystem functioning such as vegetation dynamics and carbon cycling. In cold environments, the persistence of seasonal snow cover can exert a strong control on the near-surface temperatures. However, the lack of in situ measurements of both snow cover duration and surface temperatures over high latitudes has made it difficult to estimate the spatio-temporal variability of this relationship. Here, we quantified the
fine-scale variability of winter near-surface air temperatures (+2 cm) and snow cover duration (calculated from temperature time series) using a total of 441 microclimate loggers in seven study areas across boreal and tundra landscapes in Finland during 2019–2021. We further examined the drivers behind this variation using a structural equation model and the extent to which near-surface air temperatures are buffered from free-air temperatures during winter. Our results show that while average winter near-surface temperatures stay close to 0 °C across the study domain, there are large differences in their fine-
scale variability among the study areas. Areas with large topographical variation, as well as areas with shallow snowpacks, showed the greatest variation in near-surface temperatures and in snow cover duration. In the tundra, for example, differences in minimum near-surface temperatures between study sites were close to 30 °C and topography was shown to be an important driver of this variability. In contrast, flat topography and long snow cover duration led to little spatial variation, and long periods of decoupling between near-surface and air temperatures. Quantifying and understanding the landscape-wide variation
in winter microclimates improves our ability to predict the local effects of climate change in the rapidly warming boreal and tundra regions.

## 1    Introduction

Boreal and tundra ecosystems are experiencing rapid climatic change, with average temperatures rising at 2–4 times the rate of global average temperatures in recent decades (Post et al., 2019; Rantanen et al., 2022). This macroclimatic trend (i.e.,

the overall trend in ambient air temperatures) is particularly pronounced during the winter months, and causes changes in the cryosphere that strongly feed back into the macroclimatic warming (Ruosteenoja et al., 2016; Bintanja and Andry, 2017; Bormann et al., 2018). Warmer winters have led to shorter snow cover duration and shallower snowpacks(Brown and Mote, 2009; Luomaranta et al., 2019), although increasing winter precipitation may have counteracting effects in some regions (Kellomäki et al., 2010). It is insufficiently investigated how these wintertime macroclimatic changes translate into microclimatic conditions, such as thermal conditions at and near ground surface. Near-surface temperatures have been shown to be essential for understanding for example species distributions and vegetation dynamics (Ashcroft and Gollan, 2012; Opedal et al., 2015; De Frenne et al., 2021). They also control largely ground temperatures which in turn influence the survival of wildlife (Kohler and Aanes, 2004) and ecosystem processes, such as greenhouse gas fluxes and seasonal frost (Semenchuk et al., 2016; Groffman et al., 2001; Larsen et al., 2007). However, it is not yet fully understood how near-surface temperatures vary over boreal and tundra regions as a function of air temperature and local environment, such as snow cover (De Frenne et al., 2021; Aalto et al., 2022).

Microclimates can significantly differ from the macroclimate due to local climatic processes (Aalto et al., 2017; De Frenne et al., 2021). Elevation influences microclimatic air temperatures through the atmospheric lapse rate, with temperatures typically decreasing at higher altitudes. In landscapes with strong elevational gradients, cold-air pooling is also an important driver of winter air temperatures (Daly et al., 2010). At finer scales, topography influences both microclimatic temperatures and snow cover patterns by, for example, controlling the spatial distribution of incoming solar radiation and wind drift (Barry and Blanken, 2016; Sanders-DeMott and Templer, 2017). These processes are particularly relevant in areas with no forest cover, such as open tundra. In boreal forests on the other hand, vegetation structure, such as canopy cover, controls radiation and heat fluxes within the canopy, and in turn, buffers the forest microclimate relative to ambient temperatures outside the forest (De Frenne et al., 2021). A dense forest reduces snow accumulation below the canopy by intercepting part of the snowfall (Hedstrom and Pomeroy, 1998; Koivusalo and Kokkonen, 2002). The effect of forest on snow melting is more complex as forest structure controls both incoming and outgoing radiation, meaning that the total response depends considerably on for example canopy structure, tree basal area and species composition(Ellis et al., 2011; Musselman and Pomeroy, 2017; Mazzotti et al., 2023). These processes can lead to substantial fine-scale heterogeneity in microclimatic air temperatures and distribution of snow cover, which then control variation in near-surface temperatures (Aalto et al., 2017; Sanders-DeMott and Templer, 2017). So far, the lack of empirical data has precluded quantitative assessments of these links across boreal and tundra landscapes.

In cold ecosystems, the thermal buffering between microclimate and macroclimate is most pronounced in winter, when seasonal snow cover acts as an insulator between the air and the ground and can sometimes completely separate (i.e. decouple) near-surface temperature variability from ambient air temperatures (Grundstein et al., 2005; Zhang, 2005; Aalto et al., 2018). A deep snowpack, particularly early in the winter, can decouple near-surface air temperatures from the macroclimate and keep them close to 0 °C throughout the winter, whereas a shallow snowpack or absence of snow can expose ground surfaces to large variability and extreme temperatures (Grundstein et al., 2005; Pauli et al., 2013). In cold climates, a deep snowpack has shown to increase soil respiration rates and affect vegetation growth for example by sheltering low-lying vegetation and roots from fluctuating air temperatures and erosive wind-blown ice particles that can damage the overwintering shoots(Tierney et al.,

2001; Nobrega and Grogan, 2007). On the other hand, a late-melting snowpack can keep near-surface temperatures colder than surrounding air temperatures during spring and early summer, limiting the onset of the growing season (Farbrot et al., 2011; Kankaanpää et al., 2018; Kelsey et al., 2021). However, the properties of a snowpack (e.g. depth, density, albedo) can vary considerably, both spatially and temporally, influencing the thermal conductivity of snowpack and thus its impact on ground thermal regime (Sturm et al., 1997; Domine et al., 2016). This variation is impractical to measure at fine spatial resolution over large spatial domains which complicates the assessments of fine-scale effects of snow cover on the near-surface temperatures. Furthermore, as boreal and tundra ecosystems cover a wide range of winter macroclimates, the impact and importance of snow cover on near-surface temperatures is expected to vary both regionally and from year to year.

The local implications of rapidly changing winters at high latitudes are not yet fully understood. Furthermore, the importance of different landscape characteristics on microclimates may vary between regions and from one season and winter to the next, and it is important to understand this variability and its drivers (Barry and Blanken, 2016; Aalto et al., 2022). In this study, we 1) quantify the local variability of winter near-surface temperatures and snow cover in several boreal and tundra landscapes in northern Europe, 2) analyse the landscape-scale drivers of the variation, and 3) quantify the magnitude of the buffering of surface temperatures from air temperatures. Our study design consists of a large dataset of microclimatic stations (n=441) covering different boreal and tundra landscapes in Finland. We used structural equation modelling (SEM) to investigate the hierarchical relationships among the predictors and landscape variables describing topography and vegetation. SEM is a hypothesis-driven method where a network is first constructed based on prior knowledge on how the system functions. We hypothesized air temperature to be mostly driven by coarse-scale topography while we expected snow cover duration and near-surface air temperatures to be mostly influenced by fine-scale topography as well as canopy cover. Additionally, we tested how strongly snow cover duration and near-surface air temperatures correlated with free-air temperatures and how strongly near-surface air temperatures correlated with snow cover duration. Of the two study winters (2019–2020 and 2020–2021), the first one was unusually warm, and represents conditions that are likely to become more common under climate change while the second winter was closer to average winter conditions during the normal period 1991–2020.

## 2 Material and methods

### 2.1 Study area

The study domain consists of seven focal landscapes, covering large climatic and environmental gradients from hemiboreal forests in southern Finland to oroarctic tundra in the Scandean mountains in northern Finland (Aalto et al., 2022). The macro-climate is strongly influenced by the proximity to the Arctic Ocean in the north, the Baltic Sea in the south and west, and the Eurasian continent in the east (Tikkanen, 2005). Winter conditions vary considerably across the region: average winter temperatures range from -1 °C in the southernmost study area to -13 °C in the north, while the length of the continuous snow cover period varies from three to seven months, respectively (normal period 1991–2020; Jokinen et al., 2021; Finnish Meteo-rological Insitute, 2022). The topographic relief in the study areas varies from nearly flat peatlands to areas with pronounced

topographical variation, particularly in the north. While the northernmost study areas are located in the sporadic permafrost region, permafrost is not present within any of the study areas.

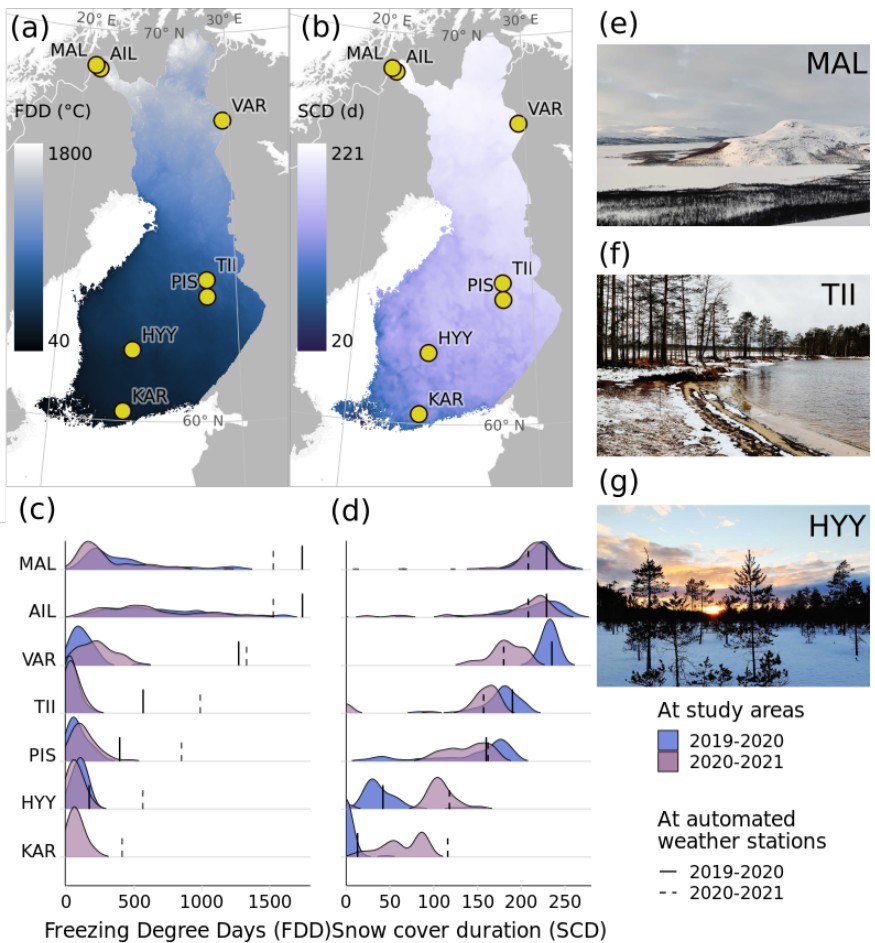

**Figure 1.** Study domain and winter temperature and snow conditions. Panels (a) and (b) represent the locations of the seven study areas in relation to (a) maximum winter freezing degree days (FDD) (1991–2020) and (b) average snow cover duration (SCD; 1991–2020) in Finland (Aalto et al., 2016). Panel (c) shows density curves of near-surface FDD calculated at the study sites within each study area during the winters (2019–2020 and 2020–2021). Panel (d) shows density curves of snow cover duration during the study period. Field photos from MAL, TII and HYY are shown in panels (e), (f) and (g). Study area abbreviations are defined in Table 1.

All seven study areas are situated in nature reserves and other protected areas in locations that represent different unmanaged environmental conditions of boreal and tundra regions (Fig. 1 and Table 1). Three of the study areas are located in northern Finland and have large elevational gradients extending below and above the tree line. Two of these, Mounts Malla (MAL) and Ailakkavaara (AIL) in Kilpisjärvi, are in the north-west, and one, Värriö Strict Nature Reserve (VAR), in the north-east. Two boreal study areas are dominated by peatlands and relatively flat topography: Tiilikkajärvi National Park (TII) in central

Finland and the Hyytiälä region, including the Siikaneva National Park in southern Finland (HYY). Another boreal study area around the Pisa Nature Reserve in central Finland (PIS) is characterized by varied topography. The Karkali Nature Reserve (KAR) lies in the southern hemiboreal zone and is surrounded by Lake Lohjanjärvi.

## 2.2 Microclimate temperature data

Each study area contains 50–100 study sites equipped with microclimate stations that continuously measure air and surface temperature throughout the year (Table 1, Fig. 4, A5 and A6). The locations of the study sites were chosen in a stratified random manner to effectively capture the main environmental gradients within each study area related to topographical variation, vegetation properties and land surface characteristics (Aalto et al., 2022). Each study site consisted of one Tomst TMS-4 logger (Wild et al., 2019), which was inserted to the soil and which measures soil temperature 6 cm below the ground and near-surface air temperatures at two heights, 2 cm and 15 cm above the surface with a precision of 0.0625 °C and an accuracy of ±0.5 °C. In this study, we described near-surface air temperatures using the measurements taken at 2 cm height which allowed us to focus on the insulating effect of snow cover and to more reliably compare surface temperatures between areas with highly varying soil properties. Additionally, air temperature (AT) at 1.5 m height was measured using either a LogTag HAXO-8 (LogTag North America Inc.; precision of 0.1 °C; ±accuracy 0.3 °C for ambient temperatures from 0 °C to 50 °C and ±0.6 °C for ambient temperatures below 0 °C) or the Onset HOBO U23 Pro v2 logger (Onset Computer Corporation; precision 0.04 °C; accuracy ±0.2 °C from 0 to 70 °C and ±0.25 from -40 to 0 °C). These loggers were placed at all study sites except in MAL and AIL, which each had 100 TMS-4 loggers and 40 air temperature loggers. The locations of these 40 loggers were chosen using a random stratified method from the locations of the original 100 sites. The AT loggers were placed under white, well-ventilated radiation shields on the north-facing side of a tree or a pole to mitigate exposure to solar radiation. Logging intervals were set to 15, 30 and 120 minutes for the TMS-4, HOBO and HAXO loggers, respectively due to varying memory and power constraints. The study period covered two winters, 2019–2020 and 2020–2021. However, not all loggers remained functional throughout the study period and thus the data from winter 2020–2021 contain fewer loggers (Table A1). The data were also checked to correct errors such as systemically too high or low temperature measurements and erroneous peaks as well as errors arising from damaged or dislocated loggers (Aalto et al., 2022).

## 2.3 Macroclimate data

Hourly weather station data, consisting of free-air temperatures measured at 2 m height and snow depth, for the study period of 01 Jan 2019 – 30 Jun 2021 and long-term averaged climate data for the most recent normal period, 1991–2020, were acquired from the nearest automated weather station (operated by the Finnish Meteorological Institute) to each study area (Finnish Meteorological Insitute, 2023). These were used to provide estimates of average macroclimatic conditions near the study areas as well as information of average snow depths. Gridded climate data (2 m air temperature and snow cover duration) were extracted from the Climgrid dataset (Aalto et al. (2016); available at https://en.ilmatieteenlaitos.fi/gridded-observations-on-aws-s3) to visualise winter air temperatures and snow cover duration throughout Finland. The dataset represents daily weather station

| | Study area | Acronym | Number of sites | Area (km$^2$) | Ecosystem |
|---|---|---|---|---|---|
| Northern Finland | Malla nature reserve | MAL | 100 (40 with AT) | 24 | Northern boreal forest & tundra |
| | Mount Ailakkavaara | AIL | 100 (40 with AT) | 24 | Northern boreal forest & tundra |
| | Värriö nature reserve | VAR | 50 | 23 | Northern boreal forest & tundra |
| Southern and Central Finland | Tiilikkajärvi national park | TII | 50 | 18 | Middle boreal forest |
| | Pisa nature reserve | PIS | 50 | 16 | Southern boreal forest |
| | Hyytiälä nature reserve | HYY | 50 | 52 | Southern boreal forest |
| | Karkali nature reserve | KAR | 50 | 48 | Hemiboreal forest |

**Table 1.** Study areas and the number of microclimate stations in each area.

observations interpolated to a 1km*1km grid using statistical interpolation with guiding variables such as topography and land cover.

## 2.4 Snow cover duration

The snow cover duration and the first and last days of the snow season were calculated for each study site based on the
variability of the near-surface temperature (+15 cm) as recorded by the TMS-4 loggers. While snow depths below 15 cm also impact near-surface temperatures, the insulating capacity of a snow pack increases with increasing depth (Zhang, 2005) and very shallow snow packs have been shown to poorly explain the relationship between free-air temperatures and near-surface air temperatures (Grundstein et al., 2005). While our estimate of the snow cover duration likely underestimates the number of days with snow on the ground, it may better reflect the insulative capacity of a snow pack by focusing on the period when snow
cover has a clear effect on near-surface temperatures.

The snow melting and arrival dates are typically straightforward to visually detect from temperature time series due to sudden changes in the magnitude of daily variation. However, these can be hard to automatically determine from the data with a simple algorithm. We created a set of rules to estimate when the loggers were under snow. Snow was estimated to be present when 1) the diurnal temperature range was less than 1 °C; 2) the maximum surface temperature stayed below 1 °C within a
centered 9-day moving window; 3) the temperature range was below 2 °C calculated with the same 9-day moving window; and lastly 4) because the moving window will slightly underestimate the snow cover duration, we tuned the snow calculations with a 5-day centered moving window where all days were deemed as snow-days if any day within the moving window was a snow-day.

We fine-tuned the temperature thresholds and moving window width by trial and error so that the determined snow arrival
and melting dates would closely match the snow period based on visual inspection of the temperature time series and to

minimize the risk of separating otherwise stable and cold conditions from snow covered periods. The deployed algorithm can be considered as conservative, and it represents the days when the snow cover is deep enough to effectively buffer the near-surface air temperatures. To validate our approach, the outcome was visually inspected for each logger (see examples in Fig. A1). However, we do not have information on the true arrival or melting dates and thus cannot provide exact evaluation statistics for the accuracy of the algorithm. Determining the snow cover duration with this method is especially challenging in situations where snow depth varies close to the height of the sensor. Nonetheless, based on our thorough visual inspections of the data, we estimate these situations to be rare in our study domain and thus the algorithm was considered reliable in detecting periods of snow cover. On average the snow cover onset and offset dates were close to those measured at the nearby weather stations (Fig. A2). The code for calculating the snow cover duration is available in the study-area-specific Github repositories (https://github.com/poniitty?tab=repositories).

## 2.5 Geospatial datasets

We utilized several open geospatial datasets to understand how different landscape characteristics affect ground thermal conditions, snow cover duration and local air temperatures. For topographical variables, we used LiDAR (light detection and ranging) data provided by the National Land Survey of Finland (NLSF; https://www.maanmittauslaitos.fi/en/maps-and-spatial-data/expert-users/product-descriptions/laser-scanning-data), collected in 2016–2019. From these data, we calculated a digital terrain model (DTM) with a resolution of 2 m for each study area using the lidR R library (Roussel et al., 2020). From the DTM, we calculated the annual sum of potential incoming solar radiation (PISR) using the Potential Incoming Solar Radiation tool in the SAGA-GIS software (version 7.6.2; http://www.saga-gis.org/saga_tool_doc/7.6.2/ta_lighting_2.html). We also calculated the Topographic Position Index (TPI) which describes the elevational difference between a grid cell and its surrounding cells using a radius of 20 m and 500 m (hereafter, TPI20 and TPI500). It was calculated using the Topographic Position Index tool in SAGA-GIS (http://www.saga-gis.org/saga_tool_doc/7.6.2/ta_morphometry_18.html). A canopy height model (resolution 1 m) based on the same LiDAR data described above was provided by the Finnish Forest Center (https://www.metsakeskus.fi/fi/avoin-metsa-ja-luontotieto/aineistot-paikkatieto-ohjelmille/paikkatietoaineistot). This was used to calculate canopy cover, defined as the proportion of minimum 2 meters high vegetation within a 5 m radius. Vegetation below 2 m was not considered due to insufficient data.

## 2.6 Statistical analyses

To analyze variation of near-surface and air temperatures between and within the study areas as well as between the two winters, we calculated three summary variables for each study site: Freezing Degree Days (FDD), mean February temperature, and minimum winter temperature. FDD was calculated as a cumulative sum throughout the snow cover period which was defined separately for each study area. Mean February temperature was used to describe average thermal conditions during winter as most of the study sites in all study areas were under snow cover during February.

We used a structural equation modelling framework (SEM) to study the direct and indirect links between the spatial variation in temperatures, snow cover, topography and canopy cover in mid-winter, and at the end of the snow cover period. SEM is

a statistical method for combining pathways of multiple predictor and response variables into a single hierarchical network

(Grace et al., 2010). We used the SEM implementation in the R package 'piecewiseSEM', version 2.1.0 (Lefcheck, 2016). In SEM, variables can appear as both predictors and responses (i.e. endogenous variables), thus allowing the investigation of indirect, mediating or cascading effects of a multivariate system (Lefcheck, 2016).

To describe air and near-surface air temperatures in the SEMs, we calculated two-week averages of both temperatures in the middle and at end of the snow cover season for each study site. The timing of the snow cover season used to calculate these

185 temperatures was defined separately for each study area. The end of the snow cover season was estimated to be when 90 % of the study sites within the study area were snow-free and the averaged temperatures were calculated from the last two weeks before the end of the period. We chose to use two weeks as the averaging period to avoid them overlapping in the southern areas. We described the snow conditions at each study site by using the total snow cover duration in the mid-season SEMs and the melting date in the late-season SEMs. We tested using snow arrival date instead of snow cover duration in the mid-winter

models but these models had mostly a lower amount of explained variance than the ones with snow cover duration, and the relationship between snow cover and near-surface air temperatures was considerably weaker (Table A2).

We expected that the strength of the links between variables may be different in the more southern and northern regions, and thus, fitted the SEMs separately for the four southernmost (KAR, HYY, PIS and TII) and three northernmost study areas (VAR, MAL and AIL). Additionally, our dataset includes data from two winters that had very different snow conditions for

many of the study areas. Thus, we decided to fit the SEMs also separately for the two winters. Therefore, two study periods, two winters and two spatial study domains resulted in eight different models.

In addition to the temperature and snow variables, we included four topographic variables (namely elevation, TPI500, TPI20 and PISR)and canopy cover. We expected solar radiation to have only a marginal effect in mid-winter as the days are short throughout the country and the angle of incoming radiation is low. Therefore we only included PISR in the SEMs for late

winter conditions. Otherwise the structure of the SEMs was the same in all eight models. Because the study sites are spatially aggregated within the seven landscapes we included the study area as a random intercept in all sub-models in each SEM. To analyze the effects of the variables in a comparative manner, we used the standardized regression coefficients. The structure of the fitted SEMs is presented alongside the numerical modelling results in Fig. 5.

To estimate the buffering of near-surface temperatures from local air temperatures, we calculated the slope of a linear

regression model between near-surface and air temperatures using a 2-week moving window throughout both winters. As the logging intervals for near-surface and air temperatures varied, we first matched the time stamps of the near-surface air temperature series with the free-air temperature series.

## 3 Results

### 3.1 Macroclimatic variation

Macroclimate and snow conditions varied between the two winters (2019–2020 and 2020–2021) and between the seven study areas (Fig. 2). The winter of 2019–2020 was generally warmer, particularly in midwinter and in southern and central Finland

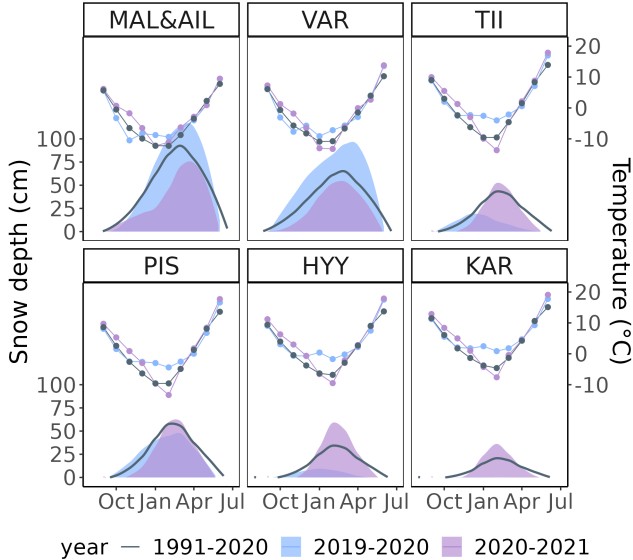

**Figure 2.** Mean macroclimatic monthly air temperature (lines) and snow depth (polygons) smoothed with local regression during winters 2019–2020 and 2020–2021, as well as during the normal period of 1991–2020, measured in the closest automated weather station to each study region (Finnish Meteorological Insitute, 2023). MAL and AIL are combined as they share the closest weather station.

(KAR, HYY, PIS and TII) with mean February temperatures nearly 10 °C warmer than during the normal period 1991–2020 and during 2020–2021. In northern Finland (VAR, AIL and MAL), temperature differences between the two winters were smaller. The winter of 2020–2021 was similar to the last normal period throughout Finland, although the autumn was generally a few degrees warmer. Snow conditions showed notable regional differences in 2019–2020. In KAR, snow cover was non-existent and there was generally little snow throughout southern and central Finland compared to the normal period. In northern Finland, on the other hand, snowpacks were thicker than usual, with snow depths exceeding 100 cm in MAL and AIL. In 2020–2021, snow depths varied less across Finland compared to 2019–2020 and each study area had several months of snow cover. While the maximum snow depth in 2020–2021 was slightly higher than average in the southern parts and slightly lower in the north, the length of the snow cover season was shorter (Figure 1).

## 3.2 Microclimatic variation

Mean near-surface air temperatures varied little between study areas during both winters compared to free-air temperatures (Fig. 3 a-b and A3). In February, mean near-surface temperatures varied from 0 °C (KAR, Feb 2020) to -4 °C (AIL, Feb 2021), while free-air temperatures varied from 1 °C (KAR, Feb 2020) to -14 °C (VAR, Feb 2021; Fig. 3 a-b). Furthermore, differences in mean near-surface temperatures within study areas (i.e. between study sites) were small throughout Finland aside from MAL and AIL where temperatures varied from 0 to -11 °C (Fig. 3 a). Compared to mean temperatures, minimum near-surface temperatures showed a larger magnitude of variation within the study areas. This variation was on average 10 °C

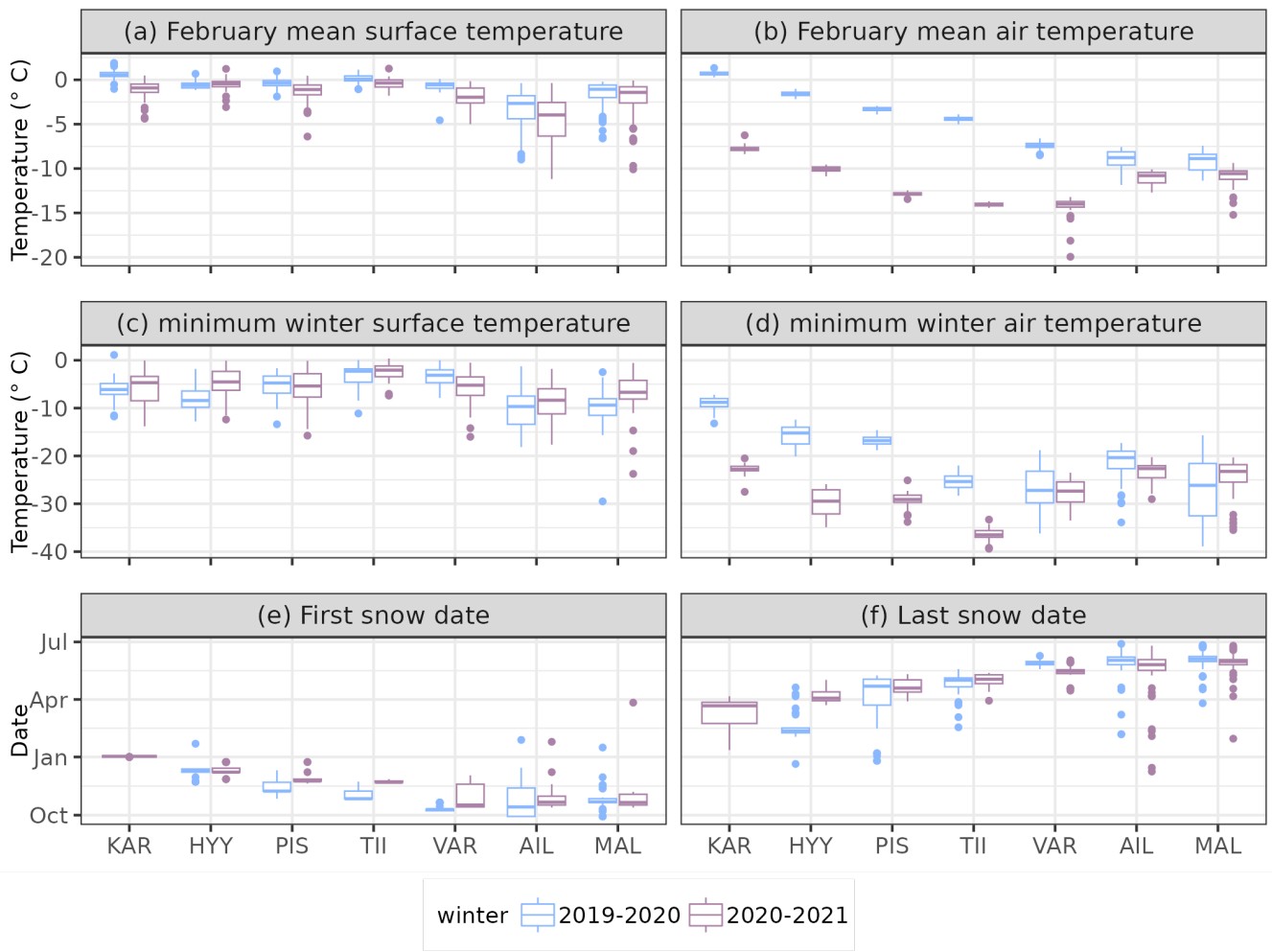

**Figure 3.** Variation in winter temperatures and snow cover in the seven study areas during two winters, 2019–2020 and 2020–2021. Figures (a) and (b) show average near-surface and free-air temperatures in February and figures (c) and (d) show minimum near-surface and air temperatures of the whole winter. Figures (e) and (f) show the onset and offset dates of snow cover. In panels (e) and (f) there is only one boxplot for KAR as there no snow was detected in KAR during winter 2019-2020.

within all study areas and 30 degrees in the northernmost study areas (Fig. 3 c). This within-area variation was also visible when looking at accumulated near-surface FDD (Figures 4, A5 and A6). In all study areas, some study sites experienced 230  nearly no freezing temperatures while others accumulated up to 550-1500 FDD in the northern study areas (Figure A5) and 200-500 FDD in the southern study areas (Figure A6). Largest spatial variation of near-surface temperatures was observed in the northernmost (AIL and MAL) study areas (Fig. 1 c-d, 4 b and A5 b) while the least within-area variation was observed in TII where nearly all microclimate loggers recorded close to 0 °C during both winters (Fig. 4 d, and Fig. A3 d). In free-air

temperatures, variance within study areas was smaller compared to near-surface temperatures while differences between study areas were larger (Fig. 3 c-d).

The duration and timing of the snow cover period also varied within and between the study areas (Fig. 4 a-c, A5, A6 and 3 e-f). Between the study areas, average snow cover duration ranged from almost nine months in the north (AIL; 2019–2020) to zero days in the south (KAR; 2019–2020) (Fig. 3 e-f). Largest within-area variations in snow cover duration were found in AIL where the duration ranged from a week to over eight months (Figure 1 and Figure 4 c) whereas for example in VAR, the difference in snow cover duration between sites was less than two months in 2019–2020 (Figure 1). Within the study areas, there was generally less variation in the snow arrival date than in the melting date (Fig. 3 e-f). First snow reached most of the study sites within each area in less than a month although in MAL and AIL in 2019–2020, the snow arrival date varied by two months across study sites (Fig. 3 e). The length of the snowmelt period was less than a month in VAR in 2019–2020 and one to five months in the other areas (Fig. 3 f).

### 3.3 Drivers of near-surface temperature variability

Structural equation models revealed season and area-specific controls for near-surface and air temperatures and snow cover duration (Fig. 5). According to Fisher's C statistics, all SEMs provided an adequate fit to the data with C ranging from 0.3 to 8.3 (p>0.05). In the northern areas, local air temperatures and snow cover duration were mostly controlled by elevation and local topography although the effects varied somewhat between seasons (Fig. 5 a-b). Elevation, for example, had a moderate positive effect (0.40, averaged over two winters) on mid-winter air temperatures but a moderate negative effect during late winter (-0.53) while fine-scale TPI (TPI 20) had a weak negative effect on snow cover duration (-0.26). In the southern areas, canopy cover had a weak negative effect on air temperature (-0.22) and a weak positive effect on SCD (0.11) (Fig. 5 c-d). In late winter, PISR had a weak negative effect (-0.26) on snow cover in both northern and southern areas (Fig. 5 b and d).

Both air temperatures and snow cover duration had a large effect in controlling near-surface temperatures. Snow cover duration had a strong positive effect (0.73) in mid-winter and a moderate negative effect (-0.49) in late winter, whereas air temperature mostly had a strong positive effect (0.81), except in mid-winter in northern areas where the effect was negative (-0.34). In mid-winter, canopy cover had a weak positive effect (0.23) on near-surface temperatures. Some effects in the models varied considerably between the two years. Most notably, local topography (both TPIs and PISR) had a significant effect on near-surface temperatures in the north only in winter 2020–2021. In southern areas, air temperatures in late winter had a positive effect (0.40) on snow cover duration in 2019–2020 but a negative effect (-0.19) in 2020–2021. The explained conditional variance, $R^2$, was generally higher in the two temperature models than in the snow cover model and in the southern areas than in the northern areas.

### 3.4 Thermal buffering due to snow

Near-surface air temperatures were largely buffered, and in many study sites decoupled (slope=0) from local free-air temperatures during winter in all study areas (Fig. 6). Insulation of near-surface air temperatures followed snow cover duration, increasing in early winter and decreasing again in late winter in all study areas. Insulation was greatest in TII, VAR and MAL

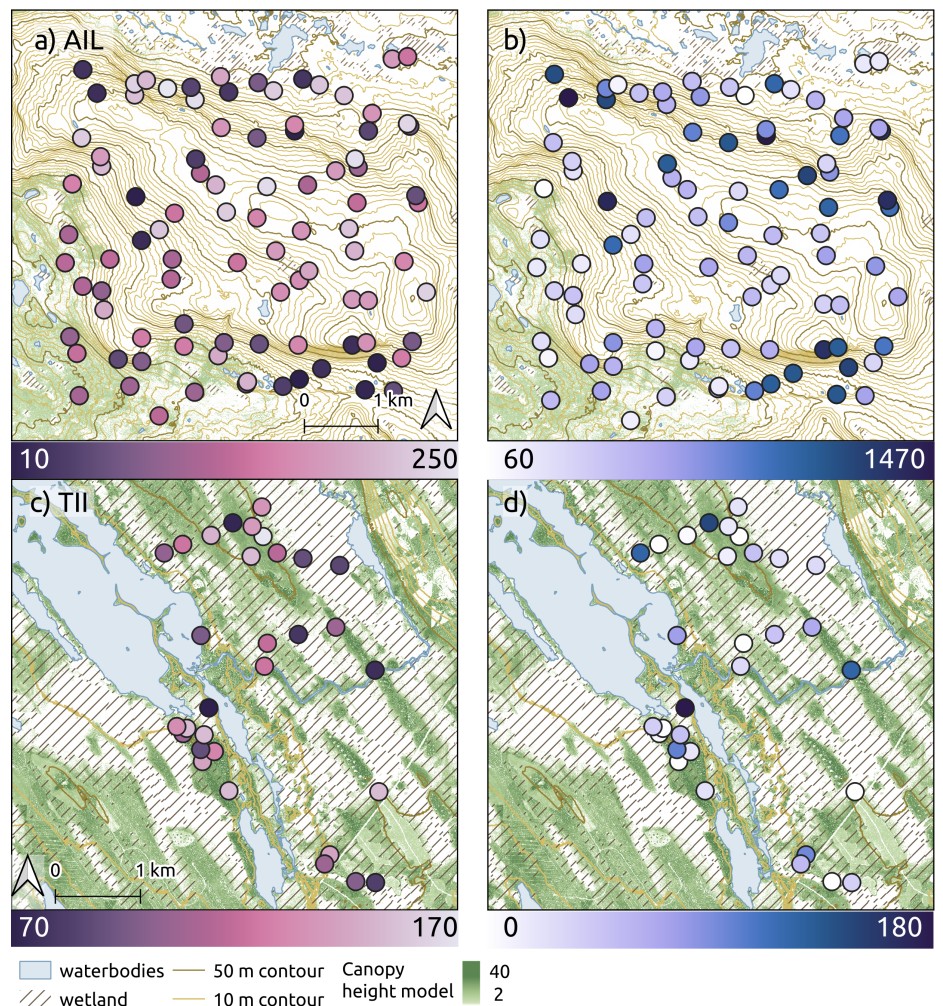

**Figure 4.** Spatial variation of snow cover duration (a and c), and Freezing Degree Days (FDD [°C days], b and d), and in AIL (a and b) and TII (c and d) based on the near-surface air temperature data during winter 2020-2021.

where the slope was close to 0.0 in nearly all study sites for several months (Fig. 6 d, c and b). In the southern study areas, temperatures were more buffered during winter 2020–2021. This was most clearly visible in HYY and PIS where the average slope was between 0.0-0.2 for most of the winter 2020–2021, whereas in 2019–2020 the slope varied both temporally and spatially across the study sites, ranging between 0.0-0.8 (Fig. 6 e and f). However, the most pronounced differences between the winters were in KAR where the complete lack of snow cover in 2019–2020 resulted in very little insulation of near-surface air temperatures (Fig. 6 g). The spatial variation within the study areas was largest in AIL where in some study sites the slope remained above 0.5 throughout the winter, particularly in 2020–2021 (Fig. 6 a). In all study areas, the insulation of near-surface

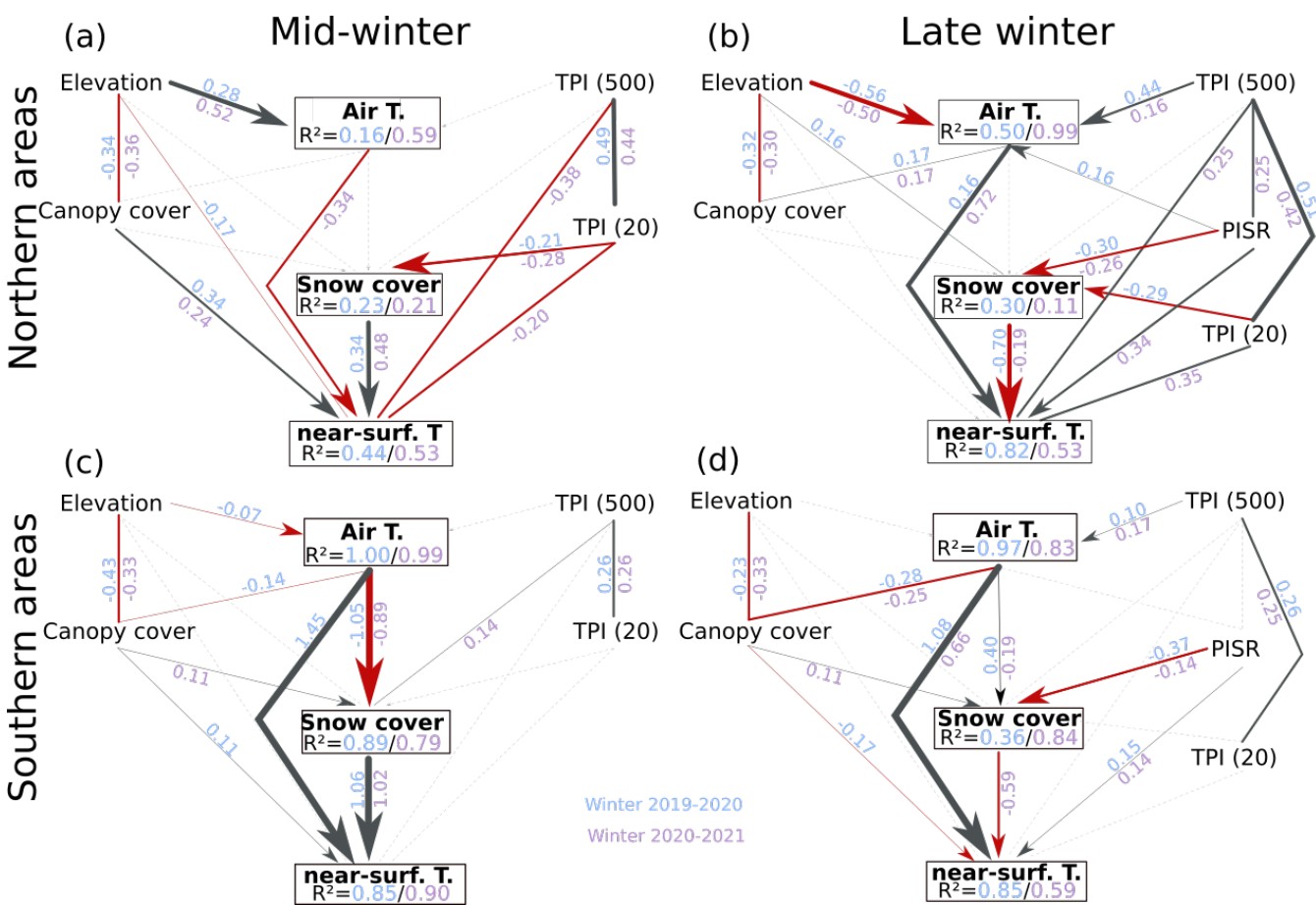

**Figure 5.** Environmental variables influencing air temperatures (shortened as Air T.) and near-surface air temperatures (near-surface T.) and snow cover duration (snow cover) in the middle of winter (a and c) and at the end of the snow cover period (b and d). Northern areas (a and b) include AIL, MAL and VAR, and southern areas (c and d) include TII, PIS, HYY and KAR. Statistically significant (P < 0.05) correlations are presented with their standardised regression slopes placed on the correlation lines. Black line indicates a positive correlation and red a negative one. Lines with no arrows indicate residual correlations. Correlations that were tested but were not found to be significant (P >0.05) are shown with light grey lines. The amount of explained variance is expressed as conditional $R^2$. TPI refers to Topographic Position Index and PISR to potential incoming solar radiation.

air temperatures was strongly related to snow cover duration and the relationship had on average a negative exponential shape
(Fig. 6 h).

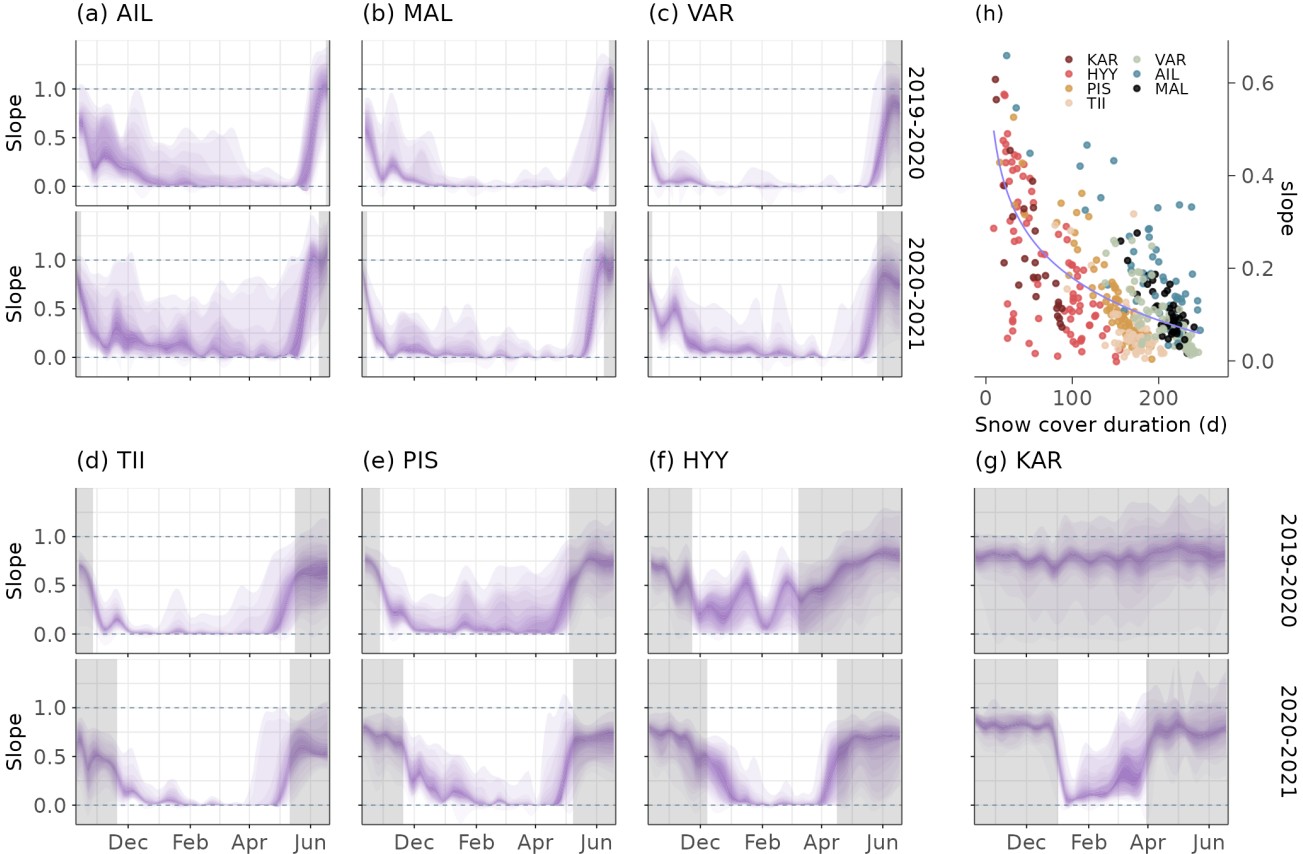

**Figure 6.** The relationship between in-situ measured near-surface and free-air temperatures in the study areas during winters 2019–2020 and 2020–2021, indicated as the slope of a linear regression model calculated from a two-week moving window. The violet color shows variation in the slope between study sites within each study area. Snow cover period for each study area is delineated with shaded grey areas. Panel (h) shows correlation and a fitted exponential function between the average length of the snow cover duration and average slope (correlation between near-surface and air temperatures) during the two winters.

## 4   Discussion

### 4.1   Winter-time heterogeneity in thermal conditions and snow cover

Our study design covered a wide gradient of winter climates and revealed considerable landscape-scale variability in microclimate temperatures and snow conditions. Air temperatures ranged from short and mild winters in the southern parts of Finland to long and severe winters in the northern tundra areas. There were notable differences between the two studied winters, as 2019–2020 was unusually warm and snow-free in the southern study areas. While average near-surface temperatures remained close to 0 °C in all study areas, the magnitude of spatial variation within the study areas varied considerably. The

largest within-area variations (up to nearly 30 °C) in near-surface air temperatures were found in the open tundra in northwestern Finland (MAL and AIL) where snow cover duration (and presumably snow depth) also varied the most. In these tundra landscapes, heterogeneous topography and low-lying vegetation strongly influence snow accumulation patterns, creating a mosaic of thermal conditions (Essery and Pomeroy, 2004; Sturm and Wagner, 2010; Gisnås et al., 2014). In contrast, in the flat, peatland-dominated TII, variation in near-surface temperatures was small, as nearly all study sites remained close to 0 °C throughout both winters. In the southern and central Finland study areas, variation in both snow cover duration and near-surface temperatures was greater in 2019–2020 than in the following year. This was likely influenced by the shallow snowpack, due to which some parts of the landscapes became snow-free early that year. This was also visible in the southernmost KAR during the winter of 2020–2021, when the area had a long-lasting but shallow snow cover.

## 4.2 Drivers of variability

The results show that microclimate temperatures and snow cover duration are influenced by topography and canopy cover. In the northern study areas, elevation and coarse-scale topography (i.e. TPI500) strongly influenced air temperatures. While the relationship between elevation and temperature is typically negative due to the atmospheric lapse rate, our results showed a positive relationship (i.e. inversion) in mid-winter, due to the strong cold-air pooling into topographic depressions and lowlands. This is in line with previous studies that have shown cold-air pooling to be a strong driver of winter temperature variability in landscapes with strong elevational gradients (Nicholas C. et al., 2009; Daly et al., 2010). Fine-scale topography (i.e. TPI20), on the other hand, had a stronger influence on snow cover duration. Topographic position has been shown to correlate well with snow accumulation patterns in mountain and tundra landscapes although the relevant radius may vary between landscapes (López-Moreno et al., 2017; Bennett et al., 2022). In addition to TPI, incoming solar radiation influenced the distribution of snow cover in late winter. While its impact on melting rates has been shown in previous studies (e.g. Cartwright et al., 2020), some studies have found its relative importance on the distribution of snow to be small compared to other topographical parameters (Schmidt et al., 2009; Revuelto et al., 2014). It is also important to notice that particularly in the open tundra, the variables included in this study explained only part of the variation in snow cover duration and near-surface air temperatures due to considerable fine-scale heterogeneity. For example, the distribution of snow is not only dependent on topography but also wind directions (Winstral and Marks, 2002).

In the southern study areas, the influence of topography was largely replaced by canopy cover which influenced both air and near-surface temperatures, as well as snow cover duration. According to our results, there was a negative correlation between canopy cover and air temperatures during both mid and late winter. Previous studies have found that forest cover tends to have a cooling effect on winter sub-canopy temperatures, particularly during the day while the effect during nights can be warming (Renaud et al., 2011; Latimer and Zuckerberg, 2017). The effect of canopy cover on snow cover duration was positive in our study, which was likely due to slower melting rates within the forests. While snow accumulates more in open areas (Hedstrom and Pomeroy, 1998; Koivusalo and Kokkonen, 2002), the effect of canopy cover on melting rates is more complex, depending on its impacts on longwave and shortwave radiation (Ellis et al., 2011). The total effect of canopy cover on snow cover duration may thus vary depending on whether the spatial variation of snow cover is dominated by accumulation patterns or by the timing

of snow melt (Mazzotti et al., 2023). Other properties of forest canopy (for example stand age, tree species, canopy structure) have also been shown to influence snow cover duration in forests (Ellis et al., 2011; Winkler and Moore, 2006; Latimer and Zuckerberg, 2017) and thus considering them may have further increased the explained variance in the models and better accounted for the role of vegetation.

In this study, we focused on investigating the effect of topography and canopy cover on winter near-surface conditions. However, other environmental variables can also be important in certain landscapes and need to be considered in future studies with more suitable datasets. For example, soil conditions, particularly soil moisture has been shown to impact temperatures near the surface within the snow pack (DOMINE et al., 2018). High soil moisture in late autumn can keep the ground warm longer during winter which also impacts surface and snow temperatures (Outcalt et al., 1990). The relatively high temperatures in TII could be explained by this as most of the study sites in TII are situated in wetlands (Fig. 4). Furthermore, while we did not consider open tundra vegetation in this study, it has been shown to strongly impact snow and near-surface temperature patterns by controlling snow properties and accumulation patterns (Essery and Pomeroy, 2004).

## 4.3 Impact of snow cover on winter near-surface temperatures

Our results show that snow cover has a consistent and strong impact on near-surface temperatures. During the winter months, the impact was positive, with longer snow cover periods leading to warmer near-surface temperatures. In nearly all study areas, near-surface air temperatures were largely decoupled from free-air temperatures, and in most study areas, mean near-surface air temperatures in February were over 10 °C higher than the corresponding free-air temperatures. Such buffering shelters low-lying vegetation from cold temperatures and influences for example winter-time soil carbon dynamics (Tierney et al., 2001; Haei et al., 2013). During the late snow cover season, there was a clear negative correlation between snow cover duration and near-surface temperatures. This is in line with previous studies which show that a late-melting snowpack can keep near-surface temperatures considerably colder from late winter to early summer (Zhang, 2005; Farbrot et al., 2011). While this effect is typically shorter and less pronounced than the warming effect in winter, it can have important implications for the onset of the growing season (Kelsey et al., 2021).

The insulating effect of a snowpack is mostly related to its depth and other properties, such as density, rather than directly to its duration (Zhang, 2005; Farbrot et al., 2011). Nonetheless, our results indicate that snow cover duration is often also related to the insulation of near-surface temperatures. This effect is likely due to snow melting occurring more slowly in places with more snow accumulation during winter. While this generally leads to longer snow cover duration, the exact relationship between snow cover duration and the insulating effect of a snow pack is more complex as melting for example within forests is also controlled by canopy cover's impact on radiation balance (Mazzotti et al., 2023). In study areas, such as MAL and VAR, where average maximum snow depths, measured at the nearest weather station of each study area, were close to or over 100 cm and which had long snow cover periods, the insulating effect was strong throughout both winters. In southern and central Finland, where average maximum snow depths ranged from 0 to 70 cm and where the snow cover period was shorter, the effect was much more spatially and temporally variable. Previous studies have shown that the variability in the insulating effect of snow cover is largest with snow depths below 100 cm (Zhang, 2005; Gisnås et al., 2014), leading to pronounced spatial

variations in the near-surface temperatures. However, even deep average snow depths do not necessarily lead to decoupling of near-surface temperatures in all parts of a landscape due to uneven snow accumulation and melting patterns driven by heterogeneous topography, as is seen in AIL as well as in previous studies in tundra regions (Farbrot et al., 2011; Gisnås et al., 2014). This heterogeneity in near-surface winter climates supports for example diversity in vegetation patterns in the tundra
(Rissanen et al.).

## 4.4 Winter microclimates in the future

The temperature increase caused by climate change is projected to be particularly strong in the winter months in Finland and other high-latitude regions (Ruosteenoja et al., 2016). In addition to increases in the average and extreme temperatures, the length of thermal winter is expected to become shorter (Ruosteenoja et al., 2020). These and the projected changes in
precipitation also affect snow conditions, although changes in precipitation are more complex and vary regionally (Luomaranta et al., 2019). However, while both free-air temperatures and snow cover are important in determining near-surface thermal conditions, temperatures at the ground surface might not directly follow their changes. Previous studies have shown that in places with seasonal snow cover, despite rising air temperatures in winter, soil and near-surface temperatures might become colder due to reduced snow cover and depth (Brown and DeGaetano, 2011). This might have important implications for the
functioning of both boreal and tundra ecosystems. The absence of snow has been shown for example to hinder understory growth (Blume-Werry et al., 2016) and control soil respiration rates also during summer (Haei et al., 2013). However, the exact response of near-surface air temperatures largely depends on the magnitude of changes in both free-air temperatures and snow cover, and can therefore show considerable regional variability (Kellomäki et al., 2010). For example, minimum temperatures in southern Finland were on average higher in 2020–2021 despite considerably colder air temperatures due to snow cover. At
the local scale, static and more stable landscape characteristics, such as topography, will continue to create fine-scale variation in near-surface temperatures and snow cover (Aalto et al., 2018). However, generally diminishing snowpacks might mean that in some parts of a landscape, snow depths are so shallow that near-surface temperatures stay coupled with air temperatures for most of the winter while in other parts of the landscape, temperatures stay decoupled. This could increase the fine-scale heterogeneity during winters in snow-covered areas. On the other hand, when snow cover is completely absent, which is likely
to become more common during future winters in the southern parts of Finland, temperature decoupling decreases drastically, as can be seen for example in KAR in the winter of 2019–2020. Even though the scope of this study was not in improving modelling techniques by incorporating these findings, these results provide important information on winter microclimatic conditions at fine spatial resolution and across a large spatial extent. We further envisage that incorporating such data into more physically-based models could considerably improve our understanding of the thermal variability and functioning of northern
ecosystems.

## 5 Conclusions

Our results highlight the notable variation in local winter near-surface temperatures across boreal and tundra landscapes. The results show pronounced spatial heterogeneity in snow cover duration and its control on winter near-surface temperatures. In general, the greatest variation in both snow cover duration and near-surface temperatures was found in the northern study areas in the tundra where pronounced topographical variability had a strong influence on the near-surface microclimate. Landscape-level microclimate variation was lowest in the flat peatland-dominated areas. The data, consisting of two contrasting winters, also revealed considerable variation in the insulation of near-surface from air temperatures depending on snow conditions. As ground thermal conditions in winter are key drivers of various ecosystem processes, these results provide important new insights into the spatio-temporal variability of winter surface microclimate across boreal and tundra ecosystems, and how these conditions may change in the future.

*Code and data availability.* The code to calculate the snow cover duration are available at the study-area-specific Github repositories (https://github.com/poniitty?tab=repositories). The measurement data is available from the corresponding author upon request.

*Author contributions.* VT designed this study and analyzed the data; PN, VT and JA designed the SEM; PN wrote the code for calculating snow cover duration and the code for the SEM; ML, JA, PN and JK designed the study setting; All authors contributed to the data collection; VT prepared the MS with contributions from all co-authors.

*Competing interests.* The authors declare that they have no conflict of interest.

*Acknowledgements.* We thank the personnel at the Kilpisjärvi Biological research station, Kevo Subarctic Research Institute, Värriö Subarctic Research Station, and Hyytiälä Forestry Field Station for their support during fieldwork. We acknowledge funding for fieldwork and equipment by the Nordenskiöld samfundet, Tiina and Antti Herlin foundation, and Maa- ja vesitekniikan tuki ry. VT would like to acknowledge Academy of Finland Grant no. 350184 (WINMET). VT and JA acknowledge funding from the Faculty of Science, University of Helsinki (project Microclim, decision 7510145). JA and ML acknowledge Academy of Finland funding (project number 342890). PN was funded by the Academy of Finland (project number 347558), the Nessling foundation, and the Kone Foundation. JK was funded by the Academy of Finland (project number 349606), and the Arctic Interactions at the University of Oulu and Academy of Finland (project number 318930, Profi 4). TR was funded by the Doctoral Program in Geosciences, University of Helsinki. JA acknowledges the Academy of Finland Flagship funding (project number 337552).

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

555  **Appendix A**

**Table A1.** Functioning loggers of near-surface temperature (ST) and air temperature (AT) per study area during the two study winters (2019–2020 and 2020–2021).

|  | ST | | | AT | | |
|---|---|---|---|---|---|---|
|  | Total (n) | 19-20 (%) | 20-21 (%) | Total (n) | 19-20 (%) | 20-21 (%) |
| AIL | 100 | 98 | 92 | 40 | 75 | 70 |
| MAL | 100 | 99 | 98 | 40 | 88 | 83 |
| VAR | 50 | 98 | 86 | 50 | 96 | 84 |
| TII | 50 | 100 | 64 | 50 | 92 | 58 |
| PIS | 50 | 98 | 98 | 50 | 92 | 90 |
| HYY | 50 | 100 | 86 | 50 | 98 | 84 |
| KAR | 50 | 92 | 84 | 50 | 92 | 66 |

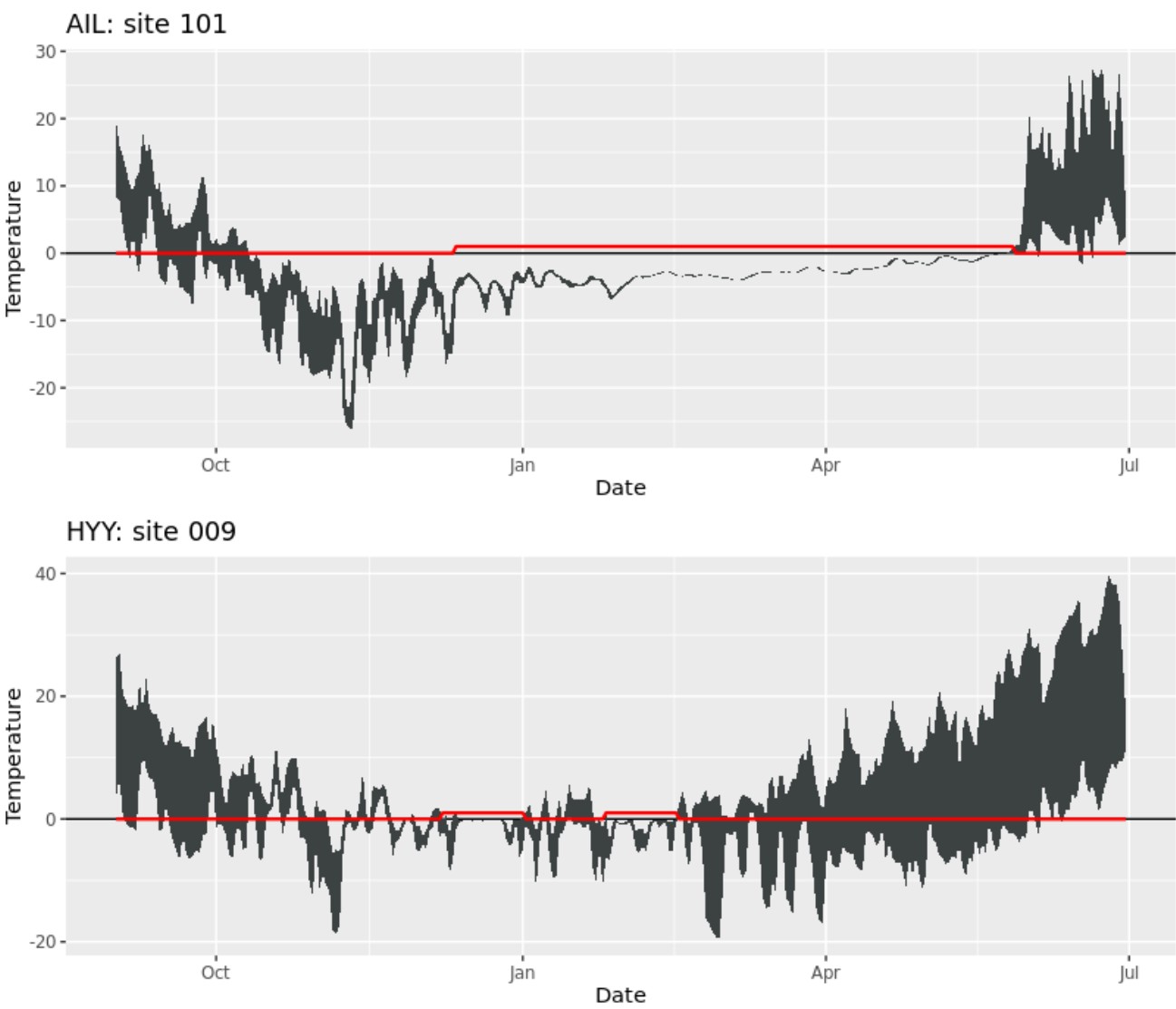

**Figure A1.** Examples of detecting snow cover duration from the temperature time series. The dark grey color shows daily maximum and minimum values of near-surface air temperatures (+15 cm) and the red line indicates when the algorithm detects snow cover (0 = no snow cover, 1 = snow cover).

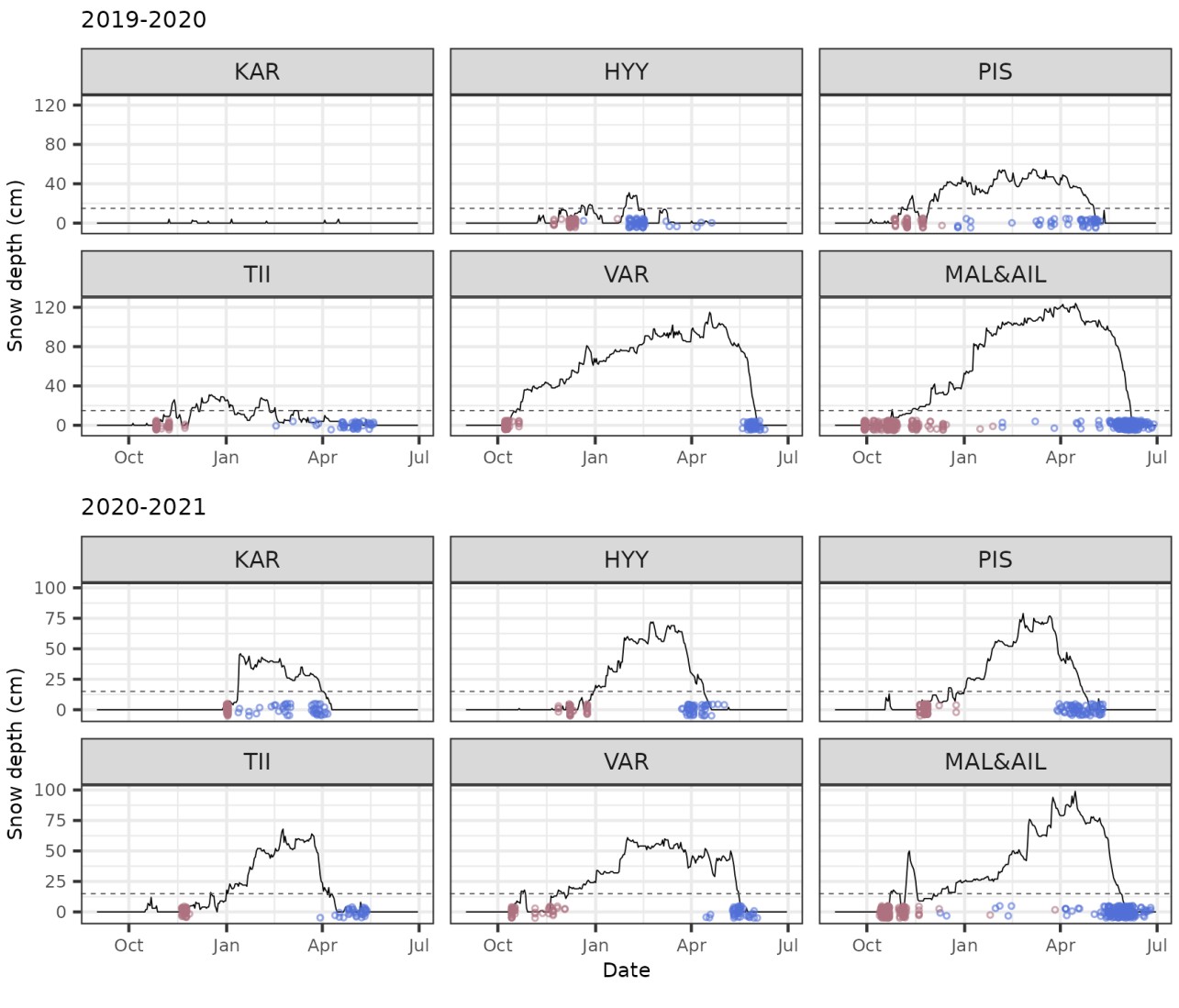

**Figure A2.** Snow cover arrival and melting dates within the study areas calculated from the near-surface air temperature time series and measured snow depth at the nearest weather station of each study area.

**Table A2.** Results of the mid-winter structural equation model using snow arrival date. Statistically significant (p <0.05) standardized effects are shown as well the total explained variance for each response variable. AT means air temperature, NST near-surface temperature and TPI Topographical Position Index.

| | Response | Predictor | Northern areas | | Southern areas | |
|---|---|---|---|---|---|---|
| | | | 2019-2020 | 2020-2021 | 2019-2020 | 2020-2021 |
| Standardized coefficients | AT | TPI (500) | - | - | - | - |
| | AT | Altitude | 0.28 | 0.52 | -0.05 | - |
| | Snow arrival | TPI (20) | - | - | - | - |
| | Snow arrival | Canopy cover | 0.18 | - | - | -0.01 |
| | Snow arrival | AT | - | - | 0.39 | -0.06 |
| | NST | AT | -0.19 | -0.28 | - | -1.83 |
| | NST | Snow arrival | - | - | -0.25 | - |
| | NST | Canopy cover | 0.31 | 0.25 | 0.2 | - |
| Residual correlations | AT | Canopy cover | - | - | -0.17 | - |
| | NST | Altitude | - | - | - | - |
| | NST | TPI (500) | -0.16 | -0.4 | - | - |
| | NST | TPI (20) | -0.16 | -0.32 | - | - |
| | Canopy cover | Altitude | -0.34 | -0.36 | -0.23 | -0.33 |
| | Snow arrival | Altitude | - | - | - | - |
| | Snow arrival | TPI (500) | -0.23 | - | - | - |
| | TPI (20) | TPI (500) | 0.49 | 0.44 | 0.27 | 0.26 |
| Explained variance (Conditional $R^2$) | AT | | 0.16 | 0.59 | 1 | 0.99 |
| | Snow arrival | | 0.37 | 0.09 | 0.19 | 1 |
| | NST | | 0.47 | 0.4 | 0.87 | 0.85 |

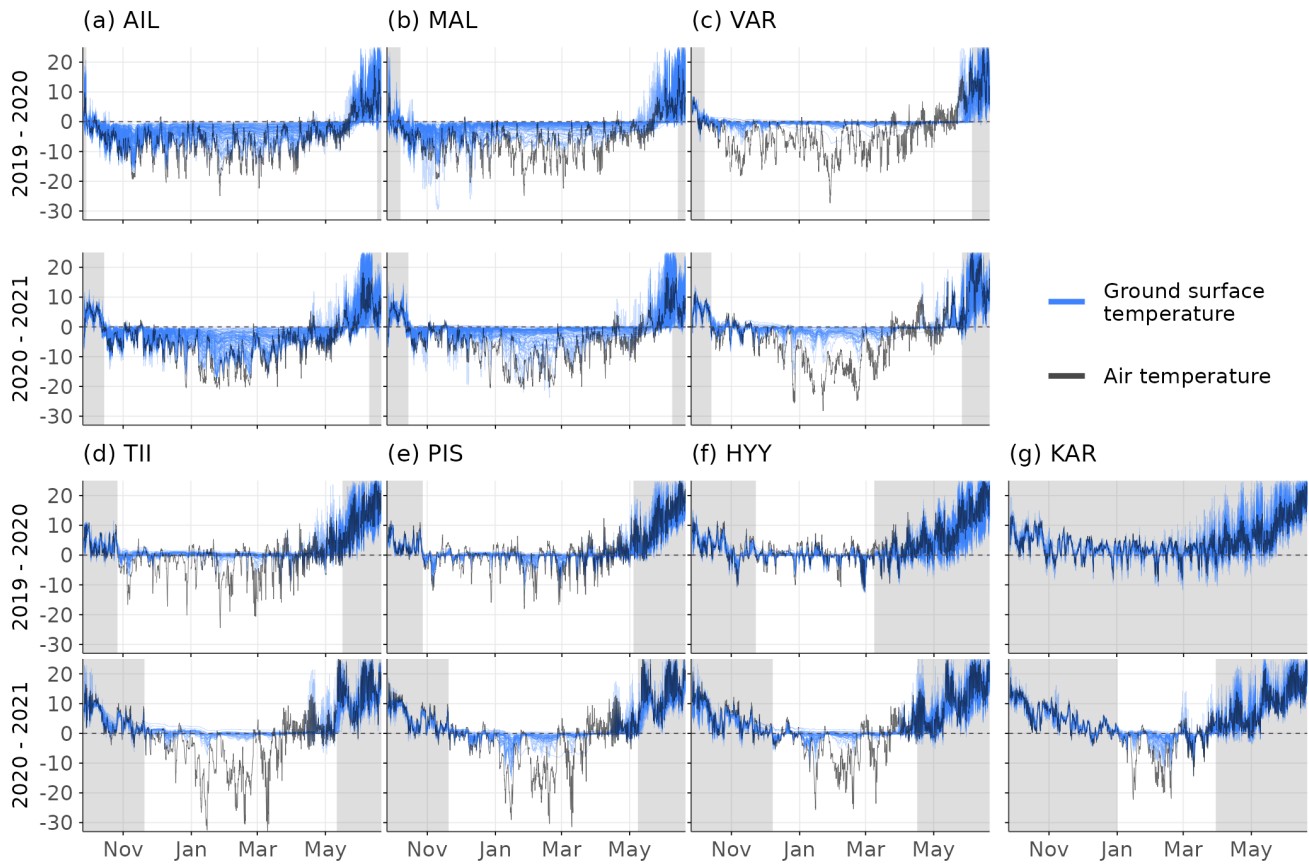

**Figure A3.** Variation of near-surface temperatures and snow cover timing during winters 2019–2020 and 2020–2021. Winter temperatures at each measurement site over the study areas are shown in the blue lines and nearest weather station temperature in the black line. The shaded areas delineate snow-free periods.

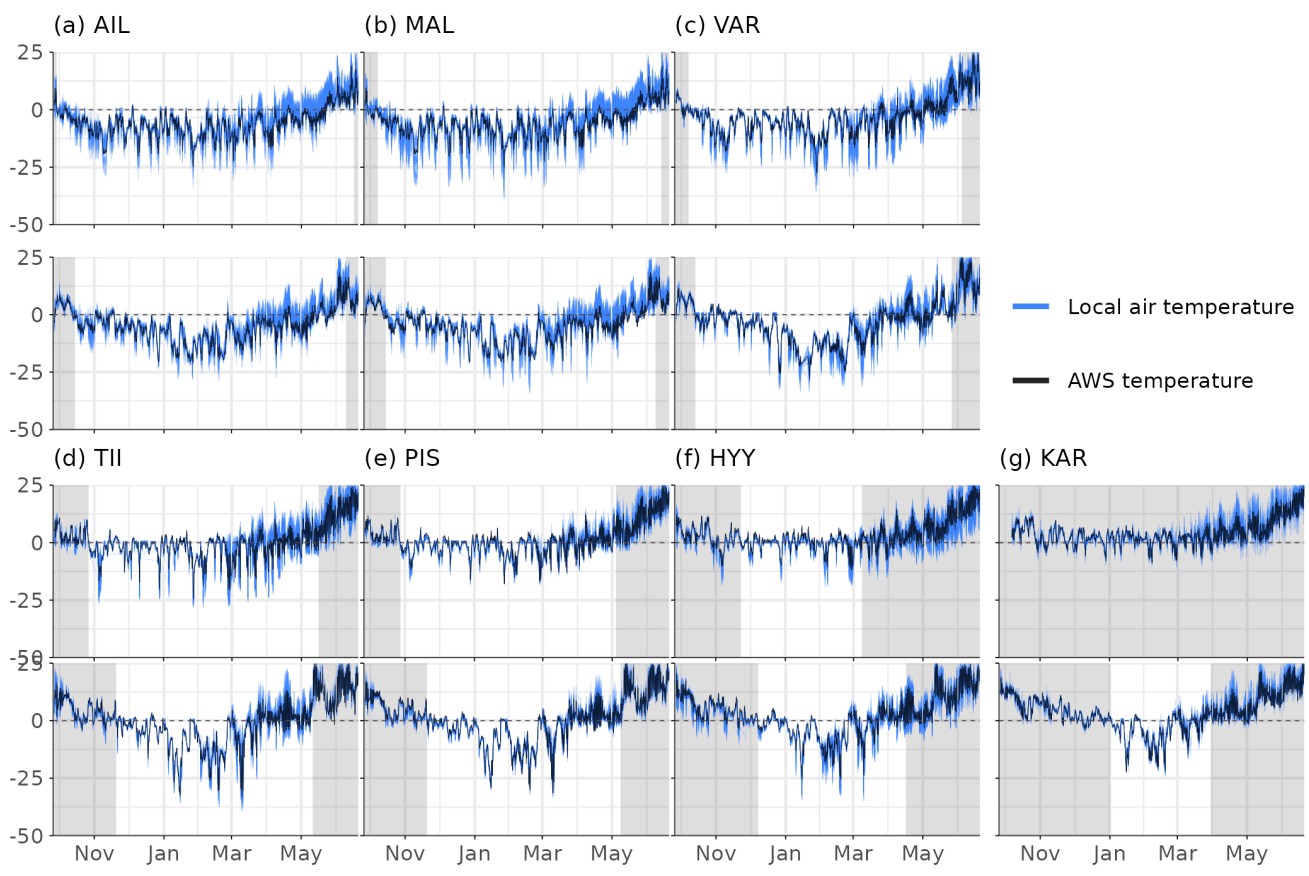

**Figure A4.** Variation of free-air temperatures and snow cover timing during the study winters 2019-2020 and 2020-2021. Winter temperatures at each study site are shown in the blue lines and nearest weather station temperature in the black line.

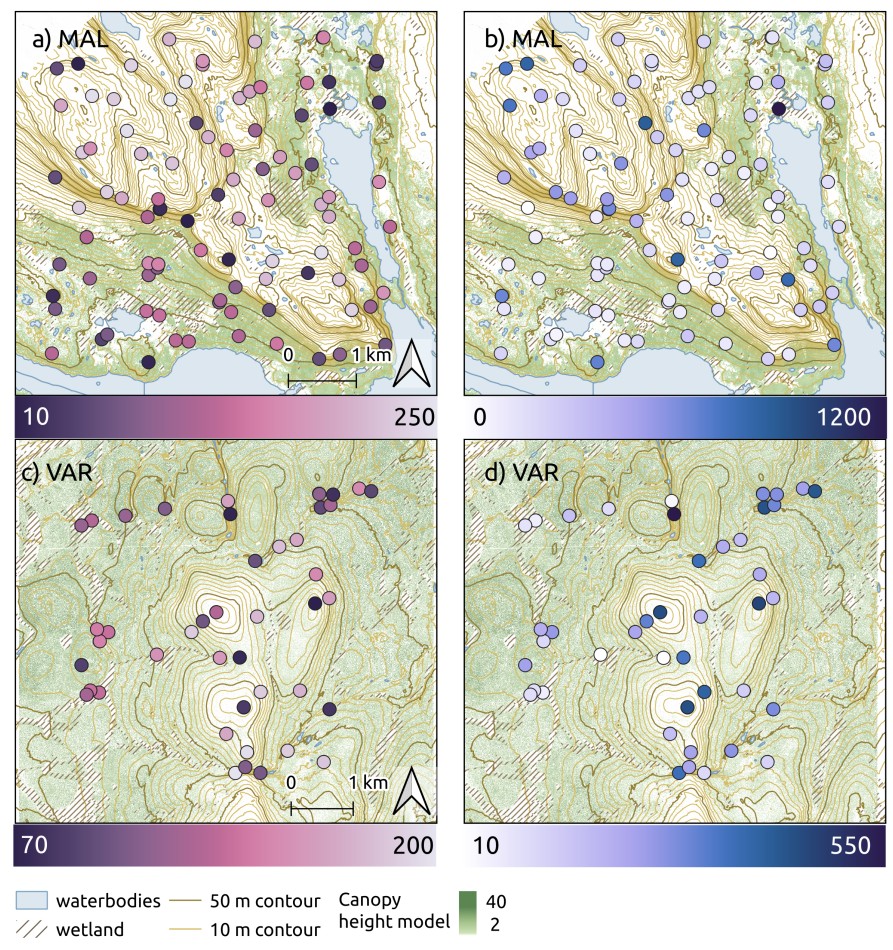

**Figure A5.** Spatial variation of snow cover duration (a and c), and Freezing Degree Days (FDD [°C days], b and d), and in MAL and VAR based on the near-surface air temperature data during winter 2020-2021.

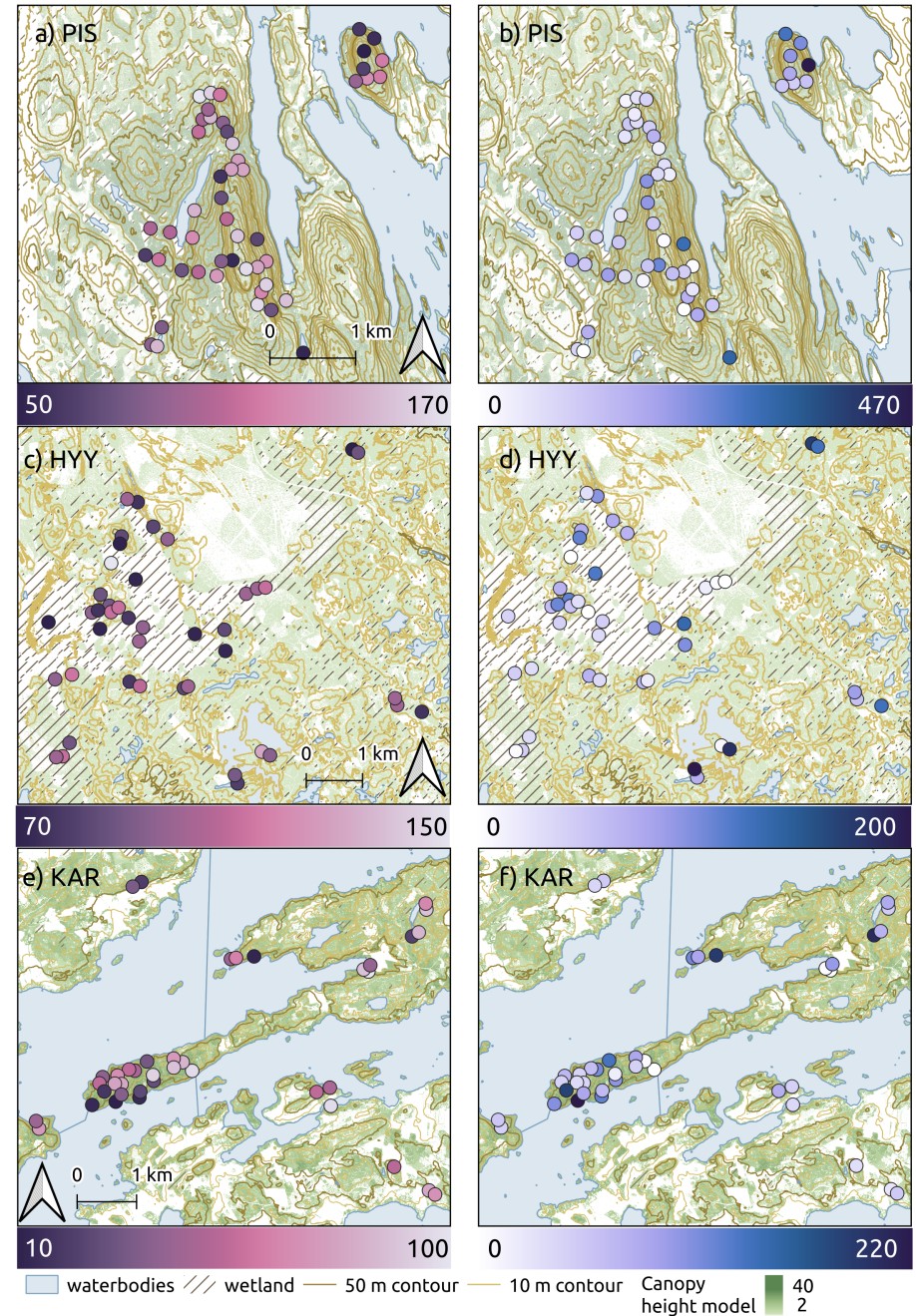

**Figure A6.** Spatial variation of snow cover duration (a and c), and Freezing Degree Days (FDD [°C days], b and d), and in PIS, HYY and KAR based on the near-surface air temperature data during winter 2020-2021.