# Peer review of "Variability and drivers of winter near-surface temperatures over boreal and tundra landscapes"

_EGUsphere, 2023_

## Author Comment (AC1)

We thank Jonathan von Oppen for his valuable and constructive feedback. His detailed and professional review was very useful and important for improving the manuscript. Here below, we provide a point-by-point response letter addressing the comments. Our responses are in blue and the line numbers (L) refer to the manuscript. The cited references are provided at the end of the letter. We thank you for your time and effort. Stay safe and take care.

On behalf of all the authors,
Sincerely,
Vilna Tyystjärvi

*General comments*

The authors investigated winter temperatures near the ground surface and how they related to topography, vegetation, and snow cover, across several boreal forest and tundra landscapes. The study addresses an important topic, as winter microclimate has long been neglected despite its importance for Arctic plant biodiversity and potential crucial carbon feedbacks from soils. While overall not surprising, the findings add important evidence on terrain-snow-temperature relationships in boreal forest and tundra.

Overall, the manuscript is well written, the study was conducted thoroughly and used considerate methodology, and the authors generally communicate their findings in an appropriate way.

> Thank you for the comment!

I do, however, see potential for improvement, as I find some methodological choices insufficiently justified and described. Despite repeatedly referring to spatial variation at the landscape level, this variation is not presented or analysed for how specific spatial patterns lay out across the landscapes studied, which could add another important insight.

> We appreciate this comment and fully agree that a more explicit presentation of the spatial variation is needed in the manuscript. We will add a figure addressing that (see a comment below regarding the same issue). However, it seems that we have somewhat differing understandings of what spatial variation means. In our opinion, variation in space, be it either within a landscape (between sites) or between landscapes (between the different study areas) constitutes as spatial variation even if this variation is not shown in a geographically explicit way. Concerning spatial patterns within the landscapes, we think that such an analysis would require a considerably denser network of microclimate stations, especially when studying winter near-ground temperatures which may vary at very fine scales and thus consider this approach outside the scope of this study.

Centrally, I suggest the authors to revisit the choice and check for consistent use of specific terminology, such as for temperature variables or study locations, to avoid confusion on the readers' side.

> We fully agree and will go through the manuscript with a fine comb to avoid further confusion.

Also, I think the impact of the study could be improved if including not only near-surface, but also below-ground temperatures, which are readily available from the dataset.

> We understand the wish for adding below-ground temperatures in the analysis and agree that understanding their variation is important. However, we don't agree that they would necessarily improve the study. We expect that they are mostly correlated with above-ground near-surface temperatures but without sufficient data concerning soil properties, we cannot adequately

investigate the possible differences between these temperatures. Furthermore, the focus of this manuscript is on near-surface air temperatures and we think that adding one more level of variation would potentially confuse rather than improve the manuscript. Nontheless, we look forward to discussing and analysing soil temperatures as well as their differences compared to air temperatures in-depth in upcoming manuscripts.

Some other shortcomings, such as the lack of in-situ snow depth data, cannot be easily overcome, but are being addressed in the manuscript. All that said, I think that the study is already presented well, and expect that these points will overall represent a minor revision effort. While some remarks concern very detailed points, that only indicates the already high level of the manuscript.

Thank you! We thoroughly appreciate the very detailed comments.

*Specific comments*

The structure of the introduction is overall logical and nice to follow, but some things are a bit confusing and should be clarified (a large part of which is related to terminology).

The methods are overall solid and thorough, but some statements lack precision and empirical support.

I find it a bit hard to extract the most important findings in the results section. There are many detailed findings being presented, and it is sometimes a bit hard to follow. I do not have a very specific suggestion to improve this though. Perhaps one point to clarify would be the specific level of variation that you are looking at, as there are at least four in parallel (sites, areas, regions, winter seasons). You could use that to structure the text (e.g., (1) across areas and within winter season, (2) within area and within winter season, (3) within area, but between winter seasons, etc).

We agree that the section 3 is currently difficult to interpret. We will go through the text in close detail and make sure that the level of variation being discussed is always explicit and the text follows a clear structure within each paragraph, addressing separately the different levels of variation.

In addition, I think it is a pity that only relationships for above-ground temperatures are being presented. Given that recent research has shown e.g. the importance of vegetation cover for winter temperatures below-ground, and that this data is also available for this study, I think it could be interesting to repeat the analyses for below-ground temperatures and include the results either in the main text or in the appendix for comparison. I appreciate that this would mean some additional work on the authors' side, but I think it would increase the scope of the study even more as it would allow conclusions with regard to winter belowground processes in boreal forest and tundra ecosystems. In my opinion, it is a key strength of the study setup with TMS loggers and elevated loggers that it enables comparison of temperatures between different heights/layers along the soil, snow and vegetation profile in the same spot, and this could be exploited even further.

As discussed above, we don't feel that with our current dataset we could draw clear conclusions concerning winter below-ground processes and therefore, we don't think that bringing soil temperatures to the study would be beneficial. We do however acknowledge that this is an important question that requires further research in the future.

Finally, I would find it interesting to see how the actual spatial temperature patterns display on the different landscapes. This could be analysed through variograms, or simply by plotting the spatial layout

of the study sites with an associated temperature variable of interest. There are repeated references to "spatial variation" in the text, but the evidence that is currently being shown only represents across-site variation irrespective of spatially explicit relationships.

> This is a good suggestion. We will add maps displaying more explicitly the spatial variation of winter near-surface temperatures in the different landscapes. We will try out the best options for implementing this but most likely we will draw maps of two to three different landscapes, showing variation in winter average near-surface temperatures as well as their temporal variation and we will add all landscapes in a supplementary figure. We will replace figure 4 with this new figure as suggested in a later comment. However, we would like to point out that the focus of our study is specifically spatial variation and not spatial patterns which in our understanding are different things, and, as explained above, inferring spatial patterns is out of our study design's scope.

Although relatively brief, the discussion covers the immediate aspects related to the study's findings well. However, I would suggest to expand a bit on the ecological implications, which are also mentioned in the first paragraph of the introduction (for instance with regard to vegetation dynamics or ecosystem processes such as permafrost dynamics or soil processes). In this context, the discussion currently only includes a brief reference to effects of snowmelt date on the start of the growing season.

> Thank you for pointing this out. We agree that ecological implications of the study could be addressed in more detail. We will improve discussion on the matter in sections 4.1 and 4.4.

*Technical corrections*

Abstract:

L2      you use "near-surface temperature" before, so it's not clear if this is the same?

> It is, we will clarify the terminology here (and throughout the manuscript).

L2f      it sounds more like you are looking at snow cover thickness, but only in L5 it becomes clear that it's about snow cover duration. I suggest to make that clear from the start.

> We will clarify this.

L10f      "In the tundra" – it's unclear if these differences were at the site/plot/… scale

> The differences were within areas, we will change the sentence: "In the tundra, for example, differences in minimum near-surface temperatures between study sites were close to 30 °C."

L11      "lead" should be "led" to match the past tense used otherwise.

L12      add a comma after "variation", else you are saying that there was little decoupling with flat topography

> Thank you for pointing these out.

*Introduction:*

L30ff     In this paragraph, I find it a bit hard to distinguish when you are talking about forest and when about tundra. I suggest re-structuring the paragraph to make it easier to follow.

> We will restructure the paragraph and and clarify which processes are relevant in boreal and tundra areas.

L45      Why are you talking about ground temperatures here, while otherwise only about near-surface temperatures? If the reason is to introduce how you deduced snow cover duration, I recommend to be explicit about it.

> We intended to mean near-surface temperatures and will fix the terminology accordingly.

L46      "absense" should probably be "absence"

> That is true.

L46f      "ground surface" is a confusing term if otherwise distinguishing between "ground temperatures" and "near-surface temperatures".

> We will harmonize the terminology used.

L49      "ice particles that affect vegetation growth" – in what way? Positively, negatively, why?

> We agree that this is a confusing sentence and will change it followingly: "In cold climates a deep snowpack has shown to affect vegetation growth for example by increasing soil respiration rates and sheltering low-lying vegetation and roots from fluctuating air temperatures and erosive ice particles that cause stress for vegetation."

Methods:

L74      Why are we looking at February specifically here?

> We included the temperature of February here as it is later used in section 3.2 but we realize that mean temperature of winter months (Dec., Jan., Feb.) might be more suitable here and will change that.

L83      Please include the abbreviation for Hyytiälä as well.

> Thank you for noticing that.

L88      How were the sites laid out in the landscape, i.e. how many plots over what size of an area? That information is essential if assessing spatial variation with regard to scale. Also, it sounds more like site locations were determined stratified randomly rather than strictly randomly, if they were aimed to cover these environmental gradients?

> The locations were indeed selected stratified randomly, we will correct that. The sizes of the study areas varied from 15 to 50 km² which definitely is essential information here. We will include the following table in the manuscript to clarify this. Related to another comment, we will also add a figure representing the spatial variation of near-surface temperatures in the study areas which will also help to clarify how the study sites were located within the landscapes.

| | Study area | Number of sites: | Area (km²) | Ecosystem |
|---|---|---|---|---|
| Northern Finland | Malla nature researve (MAL) | 100 (40 with AT) | 24 | Northern boreal forest & tundra |
| | Mount Ailakkavaara (AIL) | 100 (40 with AT) | 24 | Northern boreal forest & tundra |
| | Värriö nature reserve (VAR) | 50 | 23 | Northern boreal forest & tundra |
| Southern and Central Finland | Tiilikkajärvi (TII) | 50 | 18 | Middle boreal forest |
| | Pisa nature reserve (PIS) | 50 | 16 | Southern boreal forest |
| | Hyytiälä nature reserve (HYY) | 50 | 52 | Southern boreal forest |
| | Karkali nature reserve (KAR) | 50 | 48 | Hemiboreal forest |

L90f    Is 15 cm height really "near-surface" in tundra environments, where often much of the vegetation is below that height? If you want, you could have a look at von Oppen et al. 2022 Global Change Biology for a suggestion for terminology.

We agree that in tundra environments, other terminology such as canopy-level temperatures (as suggested in the Oppen et al. 2022 paper) would be more suitable for the 15 cm measurement height. However, considering that we also have measurements at 150 cm height and in boreal forests where 15 cm doesn't describe canopy-level temperatures, we would prefer to keep the terminology here as it is.

L97f    How did you select the 40 plots for air temperature logging out of the 100 overall plots, and how did you ensure a balanced selection?

The 40 plots were selected with another stratified random sampling from the 100 original plots in a similar way to how the 100 plots were selected. Furthermore, the selected sites and their environmental information were visually inspected to ensure that they covered all relevant gradients in the study areas. We will clarify this in the text.

L99f    Why did you choose such different logging intervals? Could that affect the data collected, e.g. underestimation of temporal variation in air T when measured through HOBO?

The loggers were set to different logging intervals due to varying memory and power constraints of the different logger types. We will clarify this in the text. We don't expect this to affect the results considering that the used air temperatures have either been averaged over sufficiently long time periods (such as figures 3 and 5) or, in the case of figure 6, only times with both near-surface and air temperature measurements were used. We will clarify the data filtering in figure 6 in the text.

L105    Maybe "weather data" should be "weather station data"? Else I don't find it intuitive that snow depth is included.

We agree and will change the term.

L105ff   The paragraph doesn't make it clear to me why both point and gridded macroclimate data were used, or if there were differences in their use.

> The gridded macroclimate data was only used in figure 1 to show variation in air temperatures and snow cover duration throughout Finland. Weather station data was used to describe winter weather conditions near the study areas in more detail in figures 2 and 1 and to get an estimate of average snow depths in the study areas during the study period. We will clarify these in section 2.3.

L112f    So snow depths of less than 15 cm were considered snow-free? I assume there would still be some insulating effect even from a thin snow layer?

> It is true that there is some insulating effect from a thin snow layer as well. However, the insulating capacity of a snowpack increases with increasing depth (see e.g. Zhang 2005) and previous studies have shown that particularly shallow autumn snowpacks poorly explain winter differences between surface and air temperatures (Grundstein 2005). Thus, we decided to focus on snow periods with over 15 cm of snow as we expected that to better describe the buffering effect of snow cover. Nonetheless, we recognize that this should be more explicitly explained in the manuscript.

L113ff   "The loggers were estimated … snow covered periods." Could you provide an empirical justification for this assumption – either from your data or citation? Why did you choose these specific moving window lengths or temperature thresholds? Also, this is a very complex and dense, yet central sentence to the paper, and I would recommend to restructure and simplify to make it easier to understand.

> The thresholds and moving window lengths were selected based on manual tests to empirically find a range that best detected the snow cover period. The outcomes were visually checked for each logger and the chosen range was considered to produce the best result for our data without underestimating or overestimating the snow cover duration. We will restructure and simplify the sentence as well as underline further the empirical nature of the algorithm and the justification for the selected parameters.

L122f    "… these situations were rare in our study domain, and the algorithm was considered to detect periods of snow cover reasonably well" – This is a very vague statement that in my opinion does not serve to increase trust in the method. Do you have e.g. in situ snow depth data that could provide empirical support?

> Unfortunately, due to limited resources we don't have in situ snow depth data to provide further empirical support for our method. We base our estimation of the quality of the algorithm on thorough manual checks for each timeseries as well as figure A2 which shows that, on average, snow departure and arrival dates align well with nearby weather station snow data. We will clarify this in the text: "While determining the snow cover duration with this method is challenging in situations where snow depth varies close to the height of the sensor, we estimate these situations to be rare in our study domain based on thorough visual inspections of the data, and the algorithm was considered to detect periods of snow cover reasonably well."

L123    "are" should perhaps be "were"? I suggest to double-check use of tenses – I prefer past tense in the methods to refer to what was done to reach the conclusions, but that might be personal preference to some degree.

We agree with the suggested tense and will carefully go through the manuscript to make sure they are used correctly.

L134    "between a point and its surroundings" – perhaps rather between a grid cell and its surrounding cells?

We agree and will change the sentence as suggested.

L139    if only incorporating vegetation > 2m, were treeless tundra sites essentially assumed to have no canopy cover?

In the scope of this study, we considered only vegetation above 2 m. This was done because the canopy data that we used was based on LiDAR data that at its current resolution is unsuitable for detecting more low-lying vegetation. We chose this dataset as it covers all of our study areas and best allowed comparisons between the landscapes which we consider to be a key strength of our study. However, we recognize that tundra vegetation plays a very important role on snow dynamics, for example controlling snow accumulation patterns (e.g. Essery & Pomeory 2004), which are not accounted for in this study. We will add further discussion about this in section 4.2 and explain in section 2.5 that vegetation below 2 m was not considered.

L141    was it really spatial variation that was assessed in SEMs? From my perspective, variation among plots and sites, yes, but perhaps not explicitly spatial?

In our understanding variation in space, ie. among different sites (and regions), is explicitly spatial variation even if the locations of the sites are not explicitly given in the model.

L147    "two-week averages" – so you only used 2 weeks out of 8 months winter data for some sites?

To describe mid-winter and late winter temperatures, we did use two-week averages. These were calculated for all of the study sites and were used only in the SEMs. We decided to use two-week averages rather than for example monthly averages to avoid overlapping between the two variables as the snow cover season especially in 2019-2020 in the southern study areas was rather short. We will explain this more accurately in the methods.

L148f   Does that mean that the end-of-snow season temperature is the two-week average before the end of snow cover season?

Yes. We will clarify that.

L149f   "Snow cover … late-season models" – As indicated above, I think "snow cover duration" would be more accurate here. Also, this sentence is quite complex and would benefit from simplifying.

We agree that in general in the manuscript, snow cover duration is a more accurate term. However, in this instance, we have particularly avoided using snow cover duration as the variable used in the late-season snow model is not snow cover duration but rather the melting date. We agree that the sentence is complex and will clarify it, explaining also better the differences in the late and mid-season models. We will also make this difference more clear in figure 5.

L152    I think the grouping approach is absolutely valid, but did you pool the data or average the effect sizes within groups of study areas?

We used the standardized regression coefficients so they could be directly compared with each other. We will clarify this in the text.

L157f    "SEM is … based on prior knowledge on how the system functions" – I think it could be useful to spell that prior knowledge out in a specific hypothesis / schematic figure etc, to clarify your expectations. Also, perhaps this descriptive bit could be combined with the background on the SEM method above (L142ff)?

> We agree that spelling out the hypotheses would be very useful as they are indeed essential for using SEMs. We hypothesized air temperature to be mostly driven by coarse-scale topography (i.e. elevation and TPI500) while we expected snow cover duration and near-surface air temperatures to be influenced mostly by more fine-scale topography (i.e. TPI20, as well as potential incoming radiation during late winter) as well as canopy cover, although we also tested the relationship with canopy cover and air temperatures. Additionally, we tested how strongly snow cover duration and near-surface temperatures correlated with air temperatures and how strongly near-surface temperatures correlated with snow cover duration. We will add these hypotheses to the manuscript, although we think they are more appropriate in section 1.

L158f    "We expected solar radiation to have only a marginal effect in mid-winter" – that is probably fair to assume, at least for the Northern study areas, but is that expectation backed up by any empirical data? Why not just include it and test this expectation?

> SEMs depend on prior knowledge of the functioning of the ecosystem. We know that there is very little sun light in mid-winter in nearly all parts of Finland and therefore consider it safe to assume that it does not have a considerable effect on either snow cover duration or near-surface air temperatures. Adding extra variables that we can reasonably expect not to have a direct effect on the predicted variables also contains a risk of misinterpreting the results. Should the model show that solar radiation had an effect in mid-winter temperatures or snow cover duration, it could also be due to another process that is related to topography. Therefore, we do not consider it useful, nor justified, to include the solar radiation variable in the mid-winter models.

L160    "similar" is too vague here in my perspective. If not identical, I suggest describing the differences in model structure explicitly.

> This is a good point. We will change the wording to "same".

L160    see my remark above on spatial arrangement of the sites. I think some background info on site distribution in each area would be helpful.

> We fully agree and will include a table and a new figure as mentioned above.

L161    "study area as a random intercept" – if I understand it correctly and "study area" = "landscape", this random intercept will not account for spatial aggregation within a study area?

> This is correct. We have accounted for the aggregation of the study sites within the different study areas, i.e. landscapes, but accounting for more explicit spatial aggregation is not, as far as we know, possible in a SEM. To avoid strong spatial aggregation within the study areas in the first place, we didn't place the study sites within 100 meters of each other when we designed the study setting.

*Results:*

L175f    I don't think the "length of the snow cover season" is actually shown anywhere explicitly, or at least it is very hard to see with the non-transparent polygons in Fig. 2, but that would be interesting and useful information. Perhaps it could be added to Fig. 3?

> We have included the length of the snow cover season in each study site in Figure 1 in the form of density curves. We will make sure it is referenced clearly in the text.

L176    "At the ground surface" vs. "near-surface" in the next sentence, but I assume they are referring to the same layer – again, I suggest you keep the terminology more consistent to avoid confusion.

> This is correct and we will go through the terminology carefully.

L176    looking at Fig. 3 a/b, there are three levels of variation that this statement could be referring to - sites, areas, and years - and if seen across sites within areas, they actually varied more (as you are also mentioning further down), so this statement is not generally true. I suggest to be explicit about which level of variation you are referring to.

> Thank you for pointing this out. In this sentence, we referred to the variation between study areas and we will change the sentence so that this is clear as well as clarify the levels of variation throughout the manuscript as explained in a previous comment.

L176    Why "mean February"? in the Methods, you only mention two-week averages. Is this referring to the same variable?

> February is in most instances the coldest month in our study areas and the month when most of the study sites are under snow cover. Our intention here was to describe average winter thermal conditions across our study areas but as the macroclimatic conditions throughout Finland vary considerably, we decided to focus on one single month rather than calculating the average temperature of all usual winter months (Dec., Jan., Feb.). The variable calculated here is, as described, mean February temperature, and does not refer to the same variable as was used in the SEM. We will clarify this variable and its use in section 2.6.

L181    "There was also more variation in winter minimum near-surface temperatures" – where was that, and more than what/where?

> We understand the confusion and will clarify the sentence: "There was also more variation within study areas in winter minimum near-surface temperatures compared to the mean February temperatures. Winter minimum temperatures varied on average by 10 degrees within the study areas and by 30 degrees in the northernmost study areas (Fig. 3 c)."

L221/223    See comments above on the use of "spatial variation" – I would use "across sites" here.

> As we have explained above, we do think that spatial variation is the correct term here. However, we will additionally specify that we refer to variation within each study area here.

L225    Perhaps it would be worthwhile mentioning the negative exponential shape of the relationship here?

> This is a good point and we will add a mention of it to the text.

*Discussion:*

L232    Either choose the term "heterogeneity" or "varied considerably" – both do not make sense here

We will correct this.

L235    "low-lying vegetation strongly influence snow accumulation patterns" – I agree from a theory point of view, but yet, SEMs did not identify a relationship between canopy cover and snow cover duration. This could be indicating the limited use of the canopy cover variable of choice for the tundra (see my remark in the methods section).

We fully agree. If we did have more accurate vegetation data, particularly describing more low-lying tundra vegetation, there might have been a more clear relationship between snow cover duration and vegetation in the tundra areas. This is a clear limitation of the vegetation variable that we selected and we will address this in more detail in the section 4.2.

L241f    It might be better to stick to "canopy cover" here for consistence (as you do below) As I see it, "vegetation structure" is more complex than the way canopy cover was measured in this study (e.g. including cover at multiple heights, maximum height etc, so essentially three-dimensional).

We agree and will modify the terminology.

L256f    I am not sure if I understand this sentence. As far as I am aware, De Frenne et al. (2019) actually found a positive buffering effect on minimum temperatures (i.e., a positive temperature offset). If this statement is meant to refer to offsets in mean temperatures, it should be rephrased to make that clearer. Also, importantly, I am in doubt if De Frenne et al. (2019) is a very appropriate reference in this context, as their analyses were mainly based on growing-season temperature records.

The sentence was trying to say that while forest cover has a positive buffering effect on minimum temperatures, the effect of forest canopy on average below-canopy temperatures in De Frenne's study was cooling. We agree that the sentence is confusing and will clarify it, and we will reference more appropriate sources. For example, Renaud et al. (2011) found that forest cover had a cooling effect on winter daytime temperatures and warming effect on night-time temperatures compared to open areas. Latimer & Zuckerberg (2017) had similar findings but also found that different characteristics of forest structure had differing impacts on winter below-canopy temperatures, highlighting the complexity of forest microclimates. We will address this complexity better in section 4.2.

L257ff   For canopy-snow interactions, you might also find the works of Malle et al (https://doi.org/10.1029/2018JD029908) and Mazzotti et al (https://doi.org/10.1029/2019WR024898, https://doi.org/10.5194/hess-2022-273) interesting, which represent some more recent developments in the field than the sources already cited.

Thank you for these sources, they provide valuable input to the discussion.

L261f    The last sentence in this paragraph does not make sense as it is now, I suggest revisiting it.

The aim of the sentence was to say that including other vegetation-related variables in addition to canopy cover may have improved the modelling results but we agree that the sentence is currently poorly written. We will modify it while we add further discussion about the role of canopy cover in controlling snow cover duration, as suggested above.

L274    Maybe reiterate for the readers that these snow depths were measured at weather stations and not in situ. With regard to spatial variability, some of the above-mentioned references might be relevant here, too.

       That is a good point, we will add a mention of the weather stations here.

*Figures:*

Fig. 1    for panel c) and d): it could be nice to have the comparison with the 1991-2020 period here as well, like in Figure 2. In the legend, it says "study areas", whereas in the text, I think these have been referred to as "sites? Please indicate the data source in the figure caption.

       Adding the normal period 1991-2020 period is a good suggestion which we will implement. We will also indicate the data source which for the panels a and b is the gridded macroclimate data mentioned in section 2.3. The photographs in panels e-g were taken by Vilna Tyystjärvi. We refer here to the study areas as the density curves show the variation within them (i.e. between the study sites in each study area). We will clarify this in the caption.

Fig. 2    Please indicate the data source in the figure caption. I find lines a bit misleading if showing monthly means for temperature, as they give the impression of continuous data. Maybe use dots instead or in addition? Adding outlines to the polygons (colour = ... argument in ggplot), or adjusting the colour scheme and/or transparency could make the snow depth data more readable. Also, I suggest to include the keyword macroclimate at the outset of the caption since that is used to refer to the figure in the text paragraph.

       We will add dots to the figure as suggested and the mention of macroclimatic data. We will make the polygons more readable in one way or another. We did try adding outlines to the polygons which didn't particularly improve the readability.

Fig. 3    I suggest to make it clear that there was (apparently?) no snow in KAR in 2020-21. I recommend to add units to the temperature axes. Also, I find it difficult to compare variation in snow cover start vs end date with different axes on panels e) vs f), I think aligning them would make it easier to verify the statements made in the text (L193f).

       There was snow in Karkal in 20-21 but not in 19-20. We will add a mention of this in the figure caption, and add units to the y-axis. Aligning the panels e and f is a good suggestion which we will implement.

Fig. 4    Aren't the shaded areas the snow cover-free periods, contrary to what it says in the caption? In addition, I noticed that this figure is actually being referred to very little in the text, and I think it adds relatively little to the information shown in Fig. 3, so you might consider moving it to the appendix.

       We agree that currently Figure 4 does not bring much information to the manuscript. We will move it to the appendix and replace it with a new figure describing the spatial variation of near-surface temperatures within the landscape more explicitly as explained in a previous comment. We will also correct the caption.

Fig. 5    I recommend explaining the variable abbreviations in the caption. That would help to make the figure more stand-alone, so readers don't have to look them up in the text. Also, I suggest renaming the response variable "surface T." to "near-surface T." to increase coherence with the text.

       We will clarify the caption and modify the figure as suggested.

Fig. 6    It is not clear from the plot if all vertical axes show beta or temperature differences. I suggest adding a label for clarification.

> All vertical axes do indeed represent beta but this should be clearer from the figure. We will clarify the axes titles.

I hope that the authors will find my remarks helpful. I wish them good luck and all the best!

Jonathan von Oppen

> Once again, thank you for your thorough comments! We also wish you all the best!

References:

Essery, R. and Pomeroy, J. (2004), Vegetation and topographic control of wind-blown snow distributions in distributed and aggregated simulations for an Arctic tundra basin, *Journal of Hydrometeorology*, 5, 735–744, doi.org/10.1175/1525-541(2004)005<0735:VATCOW>2.0.CO;2.

Grundstein, A., Todhunter, P., and Mote, T. (2005), Snowpack control over the thermal offset of air and soil temperatures in eastern North Dakota, *Geophys. Res. Lett.*, 32, L08503, doi.org/10.1029/2005GL022532.

Latimer, C.E. and Zuckerberg, B. (2017), Forest fragmentation alters winter microclimates and microrefugia in human-modified landscapes. *Ecography*, 40: 158-170. doi.org/10.1111/ecog.02551.

Renaud, V., Innes, J.L., Dobbertin, M. et al (2011), Comparison between open-site and below-canopy climatic conditions in Switzerland for different types of forests over 10 years (1998−2007). Theor Appl Climatol 105, 119–127, doi.org/10.1007/s00704-010-0361-0.

Zhang, T. (2005), Influence of the seasonal snow cover on the ground thermal regime: An overview, *Reviews of Geophysics*, 43,455, doi.org/10.1029/2004RG000157.

---

## Author Comment (AC2)

We thank the anonymous referee for their valuable and constructive feedback. Their detailed and professional review was very useful and important for improving the manuscript. Here below, we provide a point-by-point response letter addressing the comments. Our responses are in blue and the line numbers (L) refer to the manuscript. The cited references are provided at the end of the letter. We thank you for your time and effort. Stay safe and take care.
On behalf of all the authors,
Sincerely,

Vilna Tyystjärvi

The manuscript presents an analysis of winter near-surface temperatures along a south-north transect, going from boreal forest to tundra regions in Finland. The manuscript studies the drivers affecting the near-surface temperature in winter using a Structural Equation Modelling framework for the boreal and the tundra regions. Results show that snow cover duration has a strong control on soil temperature, but with opposite effect for mid-winter compared with late-winter. Site with shallow snowpack show stronger spatial variability, while site with flat topography and deep snow show strong decoupling between air temperature and soil temperature.

The manuscript is overall well written and generally show expected results. The findings of the study are not necessarily new; however, the dataset is quite extensive along a south-north gradient, which add some value to the study. The analysis is based on an interesting statistical approach which allows to see the interactions between the different drivers. However, such approach limits the applicability of the findings and in that sense a discussion on the possibility to use the dataset to improve physical modelling of snow and soil temperature could be interesting.

> Thank you for the feedback! We agree that combining this dataset with physical snow modelling would be interesting and is certainly something we have thought to research in the future. While we do not think the findings of this study directly benefit modelling in a concrete way, the possibilities of the dataset should be discussed more thoroughly. We will discuss this in more detail in section 4.

Some aspects of the methodology should also be clarifier. Overall, the manuscript is suitable for The Cryopshere, but proper improvement should be brought to the manuscript.

1. The abstract can be clarified. For example, mentioning "seven study areas across boreal and tundra landscapes", it seems that the study was made across the northern hemisphere. It is important to clarify the study extent. There should be one or two sentence on the method used to get to the results (statistical approach).- Also, the results based on snow cover duration and the SEM is not quite clear in the abstract.

Thank you for the suggestions on improving the abstract. We will clarify that the study domain covers Finland. We will add that the results are based on empirical methods, explaining shortly the approach to estimating snow cover duration as well as the statistical model that we used. We will also clarify in the abstract that the results concerning snow cover and its impact on near-surface temperatures are based on the variation of snow cover duration. We will also add what the main findings from the SEMs are.

2. Line 38: "slow down snow melt during spring through energy balance controls". It is more complex than that. There are melting related to tree radiation around the trunk. It is mentioned in the discussion. Need to be clarify here.

This is a good point. The effect of canopy on below-canopy microclimates and thus snow melt is indeed more complex and depends on, for example, canopy structure and tree species composition and local topography (Ellis et al. 2011) as well as basal area (e.g. Musselman et al. 2017). We will expand this part of the introduction.

3. Figure 1. There is a need to clarify what represent c and d. Is it a histogram of all study sites for each study area?

The panels c and d show density curves of near-surface air temperatures (c) and snow cover duration (d) in all study sites within each study area. We will clarify this in the figure caption.

4. It is important in the text to well distinguish between "study area" and "study site". Sometimes, it can get confusion. Maybe using clear acronym for each could help?

Thank you for pointing this out. We will go through the text carefully to make sure that it is always clear when we are discussing about study areas and when about study sites and what the discussed level of variation is in each sentence. However, we do not think that an acronym for each would make the text easier to understand but rather complicate it further.

5. Line 91: There is a need to clarify what "-6" means. Is it 6 centimeters under the surface. So it means that the 2 cm is above the surface? So it means that the means surface is 2 cm above the ground? Needs to be clarify. How the sensors were kept above the surface?

The sensors measure temperatures 6 cm below the ground, as well as 2 cm and 15 cm above the ground surface. When discussing near-surface temperatures, we mean near-surface air temperatures. We will clarify this focus on air rather than soil temperatures in the beginning of the manuscript and explain in more detail in section 2.2 what the measured temperatures are. The sensor is a stick-like logger which is carefully pushed to the soil to the correct depth and stays in place by itself (Wild et al. 2019).

6. Snow Cover duration: I have some doubt about the snow cover duration calculation. Why using the 15 cm? Even if the 15 cm is not cover by snow, it doesn't mean the 2 cm is not cover by snow? But the problem with using 2 cm to get the snow cover duration is that you would use the same measurements to get the snow cover duration and the impact of snow on near surface temperature. This point needs to be clarified/discussed.

This is a good point. It is true that there is some insulating effect from a thin snowpack as well. However, as the insulating capacity of a snow pack increases with increasing depth (e.g. Zhang 2005) and shallow snowpacks have been shown to poorly explain the temperature decoupling between temperatures below the snowpack and above it (Grundstein 2005), we decided to focus on periods with over 15 cm of snow as we expected that to better describe the buffering effect of a snowpack. However, we recognize that this should be more explicitly explained in the manuscript and will clarify this in section 2.4. We will further consider the limitations of this approach in section 4.3.

7. In addition it is not clear if the snow cover duration was calculated for each "study site" or each "study area". Figure A1 is confusion because it shows all the near surface temperature at 2 cm (? need to be clarify) and the snow cover duration. However, that would be interesting to show the 15 cm temperature and air temperature to see how the snow cover duration was calculated.

The snow cover duration was calculated for each study site. We will clarify that in section 2.2. In Figure A1, the bottom (top) line of the dark grey ribbon shows the daily minimum (maximum) near-surface temperatures measured at 15 cm height in the sites mentioned. We appreciate that this should be phrased more clearly in the caption. Air temperature was not used in the algorithm, so we do not think that adding it to the figure would help in understanding how the algorithm functions.

8. Line 147-150: These sentences are confusing. It seems that the calculations were done for each "study area". However, all the data is available to make the calculation at each "study sites". From these sentences, I understand that the snow cover duration is calculated for a study area, when you can calculate it at each study site. It would be very important to clarify this point and clarify how many "N" are used in the SEM.

We understand that the sentence is currently confusingly written. The temperature variables as well as the snow cover duration were indeed calculated for each study site separately even though this was not explicitly mentioned in the text. In the calculation of the temperature variables, the timing of the "mid-winter" and "late winter" was defined for each study area collectively to keep this aspect of variation similar within the study areas. We will restructure these sentences to be more accurate and easier to understand.

9. It also seems that the total snow cover duration is used as a variables in the SEM to explain the near-surface temperature in mid-winter. It seems inadequate to use a full winter snow cover duration as a variable to explain near-surface temperature in the

middle of the winter? Maybe looking at the beginning of the snow cover would make more sense?

This is a good question and we understand that this point will need further explanation. As can be seen for example in figures 3 and A2, the beginning of the snow cover season is more uniform within the study areas and does not necessarily describe the accumulation patterns within the landscape as well as the total snow cover duration or snow melting date. Our reasoning for using the total snow cover duration is that this is likely to more accurately reflect where snow does and does not accumulate in a landscape, and therefore better represents the buffering effect of snow, as the more snow accumulates in a certain place, the longer it will take for it to melt as well. As can be seen in figure 5, the total snow cover duration does indeed correlate strongly with mid-winter near-surface air temperatures. However, we understand that this effect is not always obvious and regarding the melting, the whole picture is more complex, as noted in a previous comment concerning the effect of forest structure. We will add further explanation in section 2.6 concerning our choice of variables. We will also discuss the limitations of this approach further in section 4.2. We will also repeat the mid-winter SEMs using the beginning date of the snow cover season and add these in the supplement to provide further information on the effect of the timing of the snow cover season on near-surface temperatures.

10. Would be important to mentioned if the study area are in permafrost regions. It will have a impact on the thermal regime of the soil and thus on the near-surface temperature.

This is a good point as the most northern areas of the study domain are close to permafrost regions. There is no permafrost within the actual study areas and we will clarify this in section 2.1.

11. Would be useful to give a more representative acronym for the "beta".

We will change the acronym to slope which should also describe the variable appropriately.

12. line 207: "Snow cover duration had a strong positive effect (0.73) in mid-winter". It is quite surprising to get such a strong relation when the end of snow season should not have any impact on the mid-winter soil temperature?

As we explained above, while it is true that the end of the snow cover season does not directly affect mid-winter temperatures, it does reflect snow accumulation patterns which do strongly control mid-winter conditions. As mentioned in a previous comment, we will further clarify this in section 2.6.

13. Figure 6: Should clarify what is on Y axis. Also "linear regression model calculated from a two-week moving window", add "(beta)"

The y-axis in all the panels shows beta. We will clarify this and modify the caption as suggested.

14. In the discussion, it will be important to mention the soil thermal regime. Your measurement are above the ground (2 cm) if I understood well. However, it is well known that a wet soil will stay at zero curtain longer because of the latent heat. Permafrost can also alter the soil thermal regime. Even if the measurement are not done in the soil, the author have to recognize the potential strong impact of soil on the results, which are not considered in the study.

This is an excellent point. While considering soil-related processes governing near-surface temperatures was outside the scope of this study, the soil thermal regime and its variation is important and we look forward to addressing it in more detail in a future study. Here we will, as suggested, add discussion on the impact of soil properties, including soil moisture, on near-surface air temperatures in section 4.2.

15. As mentioned earlier, the results are interesting, but not quite new. Ideally, the study would have been conducted using snow physical modelling. But I understand that it is not the scope of the work. However, should be important to relate the results to possible improvement in soil temperature modeling.

While the approach of this manuscript was empirical, we agree that it would be interesting to combine our dataset with a physical snow model. In our opinion, the most relevant finding of this manuscript, concerning snow and winter microclimate modelling, is the considerable magnitude of landscape-level variation and the need to take this variation into account in order to accurately

simulate snow cover and its impact on winter microclimates. More concrete improvements of snow modelling would, in our opinion, require a different approach than the one in this manuscript but we look forward to addressing this in future studies.

16. Figure A2: not clear what "predicted" mean in that context?

Predicted here refers to the prediction made by the snow cover algorithm. We will clarify this in the caption.

**References:**

Grundstein, A., Todhunter, P., and Mote, T. (2005), Snowpack control over the thermal offset of air and soil temperatures in eastern North Dakota, *Geophys. Res. Lett.*, 32, L08503, doi.org/10.1029 /2005GL022532.

Musselman, K. N., & Pomeroy, J. W. (2017). Estimation of needleleaf canopy and trunk temperatures and longwave contribution to melting snow. Journal of Hydrometeorology, 18(2), 555-572.

Wild, J., Kopeckỳ, M., Macek, M., Šanda, M., Jankovec, J., and Haase, T. (2019). Climate at ecologically relevant scales: A new temperature and soil moisture logger for long-term microclimate measurement, Agricultural and Forest Meteorology, 268, 40–47, https://doi.org/10.1016/j.agrformet.2018.12.018.

---

## Author Response (AR1)

We thank the two reviewers for their valuable and constructive feedback. Their detailed and professional reviews were very useful and important for improving the manuscript. Here below, we provide a point-by-point response letter addressing the comments. Our responses are in blue and the line numbers (L) refer to the track-changes version of the manuscript. The cited references are provided in the manuscript. We thank you both for your time and effort. Stay safe and take care.

On behalf of all the authors,
Sincerely,
Vilna Tyystjärvi

**Reviewer 1:**

*General comments*

The authors investigated winter temperatures near the ground surface and how they related to topography, vegetation, and snow cover, across several boreal forest and tundra landscapes. The study addresses an important topic, as winter microclimate has long been neglected despite its importance for Arctic plant biodiversity and potential crucial carbon feedbacks from soils. While overall not surprising, the findings add important evidence on terrain-snow-temperature relationships in boreal forest and tundra.

Overall, the manuscript is well written, the study was conducted thoroughly and used considerate methodology, and the authors generally communicate their findings in an appropriate way.

> Thank you for the comment!

I do, however, see potential for improvement, as I find some methodological choices insufficiently justified and described. Despite repeatedly referring to spatial variation at the landscape level, this variation is not presented or analysed for how specific spatial patterns lay out across the landscapes studied, which could add another important insight.

> We appreciate this comment and fully agree that a more explicit presentation of the spatial variation is needed in the manuscript. We added a figure addressing that (see a comment below regarding the same issue). However, it seems that we have somewhat differing understandings of what spatial variation means. In our opinion, variation in space, be it either within a landscape (between sites) or between landscapes (between the different study areas) constitutes as spatial variation even if this variation is not shown in a geographically explicit way and regardless of whether you consider its autocorrelative nature (or other structures) or not. Concerning spatial patterns within the landscapes, we think that such an analysis would require a considerably denser network of microclimate stations, especially when studying winter near-ground temperatures which may vary at very fine scales and thus consider this approach outside the scope of this study.

Centrally, I suggest the authors to revisit the choice and check for consistent use of specific terminology, such as for temperature variables or study locations, to avoid confusion on the readers' side.

> We fully agree and have gone through the manuscript with a fine comb to avoid further confusion.

Also, I think the impact of the study could be improved if including not only near-surface, but also below-ground temperatures, which are readily available from the dataset.

> We understand the wish for adding below-ground temperatures in the analysis and agree that understanding their variation is important. However, we don't agree that they would necessarily improve the study. We expect that they are mostly correlated with above-ground near-surface temperatures but without sufficient data concerning soil properties, we cannot adequately investigate the possible differences between these temperatures. Furthermore, the focus of this manuscript is on near-surface air temperatures and we think that adding one more level of variation would potentially confuse rather than improve the manuscript. Nontheless, we look forward to discussing and analysing soil temperatures as well as their differences compared to air temperatures in-depth in upcoming manuscripts.

Some other shortcomings, such as the lack of in-situ snow depth data, cannot be easily overcome, but are being addressed in the manuscript. All that said, I think that the study is already presented well, and expect that these points will overall represent a minor revision effort. While some remarks concern very detailed points, that only indicates the already high level of the manuscript.

> Thank you! We thoroughly appreciate the very detailed comments.

*Specific comments*

The structure of the introduction is overall logical and nice to follow, but some things are a bit confusing and should be clarified (a large part of which is related to terminology).

The methods are overall solid and thorough, but some statements lack precision and empirical support.

I find it a bit hard to extract the most important findings in the results section. There are many detailed findings being presented, and it is sometimes a bit hard to follow. I do not have a very specific suggestion to improve this though. Perhaps one point to clarify would be the specific level of variation that you are looking at, as there are at least four in parallel (sites, areas, regions, winter seasons). You could use that to structure the text (e.g., (1) across areas and within winter season, (2) within area and within winter season, (3) within area, but between winter seasons, etc).

> We agree that the section 3 is difficult to interpret. We have revised it and paid close attention to how different levels of variation are explained and how the paragraphs are structured.

In addition, I think it is a pity that only relationships for above-ground temperatures are being presented. Given that recent research has shown e.g. the importance of vegetation cover for winter temperatures below-ground, and that this data is also available for this study, I think it could be interesting to repeat the analyses for below-ground temperatures and include the results either in the main text or in the appendix for comparison. I appreciate that this would mean some additional work on the authors' side, but I think it would increase the scope of the study even more as it would allow conclusions with regard to winter belowground processes in boreal forest and tundra ecosystems. In my opinion, it is a key strength of the study setup with TMS loggers and elevated loggers that it enables comparison of temperatures between different heights/layers along the soil, snow and vegetation profile in the same spot, and this could be exploited even further.

> As discussed above, we don't feel that with our current dataset we could draw clear conclusions concerning winter below-ground processes and therefore, we don't think that bringing soil temperatures to the study would be beneficial. We do however acknowledge that this is an important question that requires further research in the future.

Finally, I would find it interesting to see how the actual spatial temperature patterns display on the different landscapes. This could be analysed through variograms, or simply by plotting the spatial layout of the study sites with an associated temperature variable of interest. There are repeated references to "spatial variation" in the text, but the evidence that is currently being shown only represents across-site variation irrespective of spatially explicit relationships.

> This is a good suggestion. We have added a new figure 4 which shows variation in snow cover duration and FDD in two of the study areas (AIL and TII). Similar maps of the other study areas can now be found from the appendix (figures A5 and A6). However, we would also like to point out that the focus of our study is specifically spatial variation and not spatial patterns which in our understanding are different things, and, as explained above, inferring spatial patterns is out of our study design's scope.

Although relatively brief, the discussion covers the immediate aspects related to the study's findings well. However, I would suggest to expand a bit on the ecological implications, which are also mentioned in the first paragraph of the introduction (for instance with regard to vegetation dynamics or ecosystem processes such as permafrost dynamics or soil processes). In this context, the discussion currently only includes a brief reference to effects of snowmelt date on the start of the growing season.

> Thank you for pointing this out. We agree that ecological implications of the study could be addressed in more detail. We have added discussion points on the matter in L 341 and L361-362.

*Technical corrections*

Abstract:

L2       you use "near-surface temperature" before, so it's not clear if this is the same?

> It is, we have clarified the terminology here (and throughout the manuscript).

L2f       it sounds more like you are looking at snow cover thickness, but only in L5 it becomes clear that it's about snow cover duration. I suggest to make that clear from the start.

> We  have clarified this in L2.

L10f       "In the tundra" – it's unclear if these differences were at the site/plot/… scale

> The differences were within areas, we have changed the sentence in L12: "In the tundra, for example,       differences in minimum near-surface temperatures between study sites were close to 30 °C."

L11       "lead" should be "led" to match the past tense used otherwise.

L12       add a comma after "variation", else you are saying that there was little decoupling with flat topography

> Thank you for pointing these out.

*Introduction:*

L30ff    In this paragraph, I find it a bit hard to distinguish when you are talking about forest and when about tundra. I suggest re-structuring the paragraph to make it easier to follow.

     We have clarified the paragraph in L32-47.

L45      Why are you talking about ground temperatures here, while otherwise only about near-surface temperatures? If the reason is to introduce how you deduced snow cover duration, I recommend to be explicit about it.

     We intended to mean near-surface temperatures and have fixed the terminology accordingly.

L46      "absense" should probably be "absence"

     That is true.

L46f     "ground surface" is a confusing term if otherwise distinguishing between "ground temperatures" and "near-surface temperatures".

     We have harmonized the terminology used.

L49      "ice particles that affect vegetation growth" – in what way? Positively, negatively, why?

     We agree that this is a confusing sentence and have clarified it in L54-57.

Methods:

L74      Why are we looking at February specifically here?

     We included the temperature of February here as it is later used in section 3.2 but we realize that mean temperature of winter months (Dec., Jan., Feb.) is more suitable here (L86).

L83      Please include the abbreviation for Hyytiälä as well.

     Thank you for noticing that.

L88      How were the sites laid out in the landscape, i.e. how many plots over what size of an area? That information is essential if assessing spatial variation with regard to scale. Also, it sounds more like site locations were determined stratified randomly rather than strictly randomly, if they were aimed to cover these environmental gradients?

     The locations were indeed selected stratified randomly, we have corrected that (L101). The sizes of the study areas varied from 15 to 50 km² which definitely is essential information here. We have included a new table 1 (p. 6) in the manuscript to clarify this. Related to another comment, we have also added new figures representing the spatial variation of near-surface temperatures in the study areas which will also help to clarify how the study sites were located within the landscapes.

L90f     Is 15 cm height really "near-surface" in tundra environments, where often much of the vegetation is below that height? If you want, you could have a look at von Oppen et al. 2022 Global Change Biology for a suggestion for terminology.

     We agree that in tundra environments, other terminology such as canopy-level temperatures (as suggested in the Oppen et al. 2022 paper) would be more suitable for the 15 cm measurement height. However, considering that we also have measurements at 150 cm height and in boreal

forests where 15 cm doesn't describe canopy-level temperatures, we would prefer to keep the terminology here as it is.

L97f     How did you select the 40 plots for air temperature logging out of the 100 overall plots, and how did you ensure a balanced selection?

The 40 plots were selected with another stratified random sampling from the 100 original plots in a similar way to how the 100 plots were selected. Furthermore, the selected sites and their environmental information were visually inspected to ensure that they covered all relevant gradients in the study areas. We have clarified this (L113).

L99f     Why did you choose such different logging intervals? Could that affect the data collected, e.g. underestimation of temporal variation in air T when measured through HOBO?

The loggers were set to different logging intervals due to varying memory and power constraints of the different logger types. We have clarified this (L116). We don't expect this to affect the results considering that the used air temperatures have either been averaged over sufficiently long time periods (such as figures 3 and 5) or, in the case of figure 6, only times with both near-surface and air temperature measurements were used. We have clarified the data filtering in figure 6 (L210-211).

L105     Maybe "weather data" should be "weather station data"? Else I don't find it intuitive that snow depth is included.

We agree and have changed the term (L122).

L105ff   The paragraph doesn't make it clear to me why both point and gridded macroclimate data were used, or if there were differences in their use.

The gridded macroclimate data was only used in figure 1 to show variation in air temperatures and snow cover duration throughout Finland. Weather station data was used to describe winter weather conditions near the study areas in more detail in figures 2 and 1 and to get an estimate of average snow depths in the study areas during the study period. We have clarified these (L125-126 and L128).

L112f    So snow depths of less than 15 cm were considered snow-free? I assume there would still be some insulating effect even from a thin snow layer?

It is true that there is some insulating effect from a thin snow layer as well. However, the insulating capacity of a snowpack increases with increasing depth (see e.g. Zhang 2005) and previous studies have shown that particularly shallow autumn snowpacks poorly explain winter differences between surface and air temperatures (Grundstein 2005). Thus, we decided to focus on snow periods with over 15 cm of snow as we expected that to better describe the buffering effect of snow cover. Nonetheless, we recognize that this should be more explicitly explained in the manuscript and have now clarified this (L133-138).

L113ff   "The loggers were estimated … snow covered periods." Could you provide an empirical justification for this assumption – either from your data or citation? Why did you choose these specific moving window lengths or temperature thresholds? Also, this is a very complex and dense, yet central sentence to the paper, and I would recommend to restructure and simplify to make it easier to understand.

The thresholds and moving window lengths were selected based on manual tests to empirically find a range that best detected the snow cover period. The outcomes were visually checked for each logger and the chosen range was considered to produce the best result for our data without underestimating or overestimating the snow cover duration. We have restructured and simplified the sentence (L141-146) as well as underlined further the empirical nature of the algorithm and the justification for the selected parameters (L147-153).

L122f  "… these situations were rare in our study domain, and the algorithm was considered to detect periods of snow cover reasonably well" – This is a very vague statement that in my opinion does not serve to increase trust in the method. Do you have e.g. in situ snow depth data that could provide empirical support?

Unfortunately, due to limited resources we don't have in situ snow depth data to provide further empirical support for our method. We base our estimation of the quality of the algorithm on thorough manual checks for each timeseries as well as figure A2 which shows that, on average, snow departure and arrival dates align well with nearby weather station snow data. We have further explained this (L153-156).

L123   "are" should perhaps be "were"? I suggest to double-check use of tenses – I prefer past tense in the methods to refer to what was done to reach the conclusions, but that might be personal preference to some degree.

We agree with the suggested tense and have carefully gone through the manuscript to make sure they are used correctly.

L134   "between a point and its surroundings" – perhaps rather between a grid cell and its surrounding cells?

We agree and have changed the sentence as suggested (L167).

L139   if only incorporating vegetation > 2m, were treeless tundra sites essentially assumed to have no canopy cover?

In the scope of this study, we considered only vegetation above 2 m. This was done because the canopy data that we used was based on LiDAR data that at its current resolution is unsuitable for detecting more low-lying vegetation. We chose this dataset as it covers all of our study areas and best allowed comparisons between the landscapes which we consider to be a key strength of our study. However, we recognize that tundra vegetation plays a very important role on snow dynamics, for example controlling snow accumulation patterns (e.g. Essery & Pomeory 2004), which are not accounted for in this study. We have added further discussion about this (L333-335) and explained that vegetation below 2 m was not considered (L173).

L141   was it really spatial variation that was assessed in SEMs? From my perspective, variation among plots and sites, yes, but perhaps not explicitly spatial?

In our understanding variation in space, ie. among different sites (and regions), is explicitly spatial variation even if the locations of the sites are not explicitly given in the model.

L147   "two-week averages" – so you only used 2 weeks out of 8 months winter data for some sites?

To describe mid-winter and late winter temperatures, we did use two-week averages. These were calculated for all of the study sites and were used only in the SEMs. We decided to use two-week

averages rather than for example monthly averages to avoid overlapping between the two variables as the snow cover season especially in 2019-2020 in the southern study areas was rather short. We have explained these calculations in more detail (L186-191).

L148f Does that mean that the end-of-snow season temperature is the two-week average before the end of snow cover season?

Yes. We have clarified that (L190).

L149f "Snow cover … late-season models" – As indicated above, I think "snow cover duration" would be more accurate here. Also, this sentence is quite complex and would benefit from simplifying.

We agree and have simplified and clarified this sentence (L191-192).

L152 I think the grouping approach is absolutely valid, but did you pool the data or average the effect sizes within groups of study areas?

We used the standardized regression coefficients so they could be directly compared with each other. We have clarified this in the text (L206).

L157f "SEM is … based on prior knowledge on how the system functions" – I think it could be useful to spell that prior knowledge out in a specific hypothesis / schematic figure etc, to clarify your expectations. Also, perhaps this descriptive bit could be combined with the background on the SEM method above (L142ff)?

We agree and have added these hypotheses to the manuscript, although we think they are more appropriate in section 1 (L73-77).

L158f "We expected solar radiation to have only a marginal effect in mid-winter" – that is probably fair to assume, at least for the Northern study areas, but is that expectation backed up by any empirical data? Why not just include it and test this expectation?

SEMs depend on prior knowledge of the functioning of the ecosystem. We know that there is very little sun light in mid-winter in nearly all parts of Finland and therefore consider it safe to assume that it does not have a considerable effect on either snow cover duration or near-surface air temperatures. Adding extra variables that we can reasonably expect not to have a direct effect on the predicted variables also contains a risk of misinterpreting the results. Should the model show that solar radiation had an effect in mid-winter temperatures or snow cover duration, it could also be due to another process that is related to topography. Therefore, we do not consider it useful, nor justified, to include the solar radiation variable in the mid-winter models.

L160 "similar" is too vague here in my perspective. If not identical, I suggest describing the differences in model structure explicitly.

This is a good point. We have changed the wording to "same" (L204).

L160 see my remark above on spatial arrangement of the sites. I think some background info on site distribution in each area would be helpful.

We fully agree and have included a table (table 1) and a new figure 4 as mentioned above.

L161 "study area as a random intercept" – if I understand it correctly and "study area" = "landscape", this random intercept will not account for spatial aggregation within a study area?

This is correct. We have accounted for the aggregation of the study sites within the different study areas, i.e. landscapes, but accounting for more explicit spatial aggregation is not, as far as we know, possible in a SEM. To avoid strong spatial aggregation within the study areas in the first place, we didn't place the study sites within 100 meters of each other when we designed the study setting.

*Results:*

L175f    I don't think the "length of the snow cover season" is actually shown anywhere explicitly, or at least it is very hard to see with the non-transparent polygons in Fig. 2, but that would be interesting and useful information. Perhaps it could be added to Fig. 3?

We have included the length of the snow cover season in each study site in Figure 1 in the form of density curves. We have made sure it is referenced clearly in the text (e.g. L224).

L176     "At the ground surface" vs. "near-surface" in the next sentence, but I assume they are referring to the same layer – again, I suggest you keep the terminology more consistent to avoid confusion.

This is correct and we have cone through the terminology carefully.

L176     looking at Fig. 3 a/b, there are three levels of variation that this statement could be referring to - sites, areas, and years - and if seen across sites within areas, they actually varied more (as you are also mentioning further down), so this statement is not generally true. I suggest to be explicit about which level of variation you are referring to.

Thank you for pointing this out. In this sentence, we referred to the variation between study areas and have changed the sentence so that this is clear as well as clarified the levels of variation throughout the manuscript as explained in a previous comment.

L176     Why "mean February"? in the Methods, you only mention two-week averages. Is this referring to the same variable?

February is in most instances the coldest month in our study areas and the month when most of the study sites are under snow cover. Our intention here was to describe average winter thermal conditions across our study areas but as the macroclimatic conditions throughout Finland vary considerably, we decided to focus on one single month rather than calculating the average temperature of all usual winter months (Dec., Jan., Feb.). The variable calculated here is, as described, mean February temperature, and does not refer to the same variable as was used in the SEM. We have clarified this variable and its use (L178).

L181     "There was also more variation in winter minimum near-surface temperatures" – where was that, and more than what/where?

We understand the confusion have clarified the sentence (L231).

L221/223        See comments above on the use of "spatial variation" – I would use "across sites" here.

As we have explained above, we do think that spatial variation is the correct term here. However, we have now specified that we refer to variation within each study area here (L276).

L225     Perhaps it would be worthwhile mentioning the negative exponential shape of the relationship here?

*This is a good point and we have added a mention of it (L281).*

*Discussion:*

L232    Either choose the term "heterogeneity" or "varied considerably" – both do not make sense here

*We have corrected this (L288).*

L235    "low-lying vegetation strongly influence snow accumulation patterns" – I agree from a theory point of view, but yet, SEMs did not identify a relationship between canopy cover and snow cover duration. This could be indicating the limited use of the canopy cover variable of choice for the tundra (see my remark in the methods section).

*We fully agree. If we did have more accurate vegetation data, particularly describing more low-lying tundra vegetation, there might have been a more clear relationship between snow cover duration and vegetation in the tundra areas. This is a clear limitation of the vegetation variable that we selected and we have addressed this in more detail (L333-335).*

L241f    It might be better to stick to "canopy cover" here for consistence (as you do below) As I see it, "vegetation structure" is more complex than the way canopy cover was measured in this study (e.g. including cover at multiple heights, maximum height etc, so essentially three-dimensional).

*We agree and have modified the terminology (L299).*

L256f    I am not sure if I understand this sentence. As far as I am aware, De Frenne et al. (2019) actually found a positive buffering effect on minimum temperatures (i.e., a positive temperature offset). If this statement is meant to refer to offsets in mean temperatures, it should be rephrased to make that clearer. Also, importantly, I am in doubt if De Frenne et al. (2019) is a very appropriate reference in this context, as their analyses were mainly based on growing-season temperature records.

*The sentence was trying to say that while forest cover has a positive buffering effect on minimum temperatures, the effect of forest canopy on average below-canopy temperatures in De Frenne's study was cooling. We agree that the sentence is confusing and have clarified it as well as included more appropriate sources (L316-318).*

L257ff   For canopy-snow interactions, you might also find the works of Malle et al (https://doi.org/10.1029/2018JD029908) and Mazzotti et al (https://doi.org/10.1029/2019WR024898, https://doi.org/10.5194/hess-2022-273) interesting, which represent some more recent developments in the field than the sources already cited.

*Thank you for these sources, they have provided valuable input to the discussion (L321-323).*

L261f    The last sentence in this paragraph does not make sense as it is now, I suggest revisiting it.

*The aim of the sentence was to say that including other vegetation-related variables in addition to canopy cover may have improved the modelling results but we agree that the sentence was poorly written. We have modified it while adding further discussion about the role of canopy cover in controlling snow cover duration, as suggested above (L320-326).*

L274    Maybe reiterate for the readers that these snow depths were measured at weather stations and not in situ. With regard to spatial variability, some of the above-mentioned references might be relevant here, too.

> That is a good point, we have added a mention of the weather stations here (L353).

*Figures:*

Fig. 1    for panel c) and d): it could be nice to have the comparison with the 1991-2020 period here as well, like in Figure 2. In the legend, it says "study areas", whereas in the text, I think these have been referred to as "sites? Please indicate the data source in the figure caption.

> Adding the normal period 1991-2020 period is a good suggestion. However, we have calculated FDD across the snow cover season in each study area which means that we cannot calculate FDD during the normal period in a comparable way and thus decided not to include it. Furthermore, we feel that panels c and d might have become too crowded if adding one more small vertical line. We have indicated the data source which for the panels a and b is the gridded macroclimate data mentioned in section 2.3. The photographs in panels e-g were taken by Vilna Tyystjärvi. We refer here to the study areas as the density curves show the variation within them (i.e. between the study sites in each study area). We have clarified this in the caption.

Fig. 2    Please indicate the data source in the figure caption. I find lines a bit misleading if showing monthly means for temperature, as they give the impression of continuous data. Maybe use dots instead or in addition? Adding outlines to the polygons (colour = ... argument in ggplot), or adjusting the colour scheme and/or transparency could make the snow depth data more readable. Also, I suggest to include the keyword macroclimate at the outset of the caption since that is used to refer to the figure in the text paragraph.

> We have added dots to the figure as suggested and the mention of macroclimatic data. We have modified the polygons by removing filling from the normal period as this seemed the only reasonable way to make the figure more easy to read.

Fig. 3    I suggest to make it clear that there was (apparently?) no snow in KAR in 2020-21. I recommend to add units to the temperature axes. Also, I find it difficult to compare variation in snow cover start vs end date with different axes on panels e) vs f), I think aligning them would make it easier to verify the statements made in the text (L193f).

> There was snow in Karkal in 20-21 but indeed not in 19-20. We have mentioned this in the figure caption, and added units to the y-axis. Aligning the panels e and f was a good suggestion which we have also done.

Fig. 4    Aren't the shaded areas the snow cover-free periods, contrary to what it says in the caption? In addition, I noticed that this figure is actually being referred to very little in the text, and I think it adds relatively little to the information shown in Fig. 3, so you might consider moving it to the appendix.

> We agree that currently Figure 4 does not bring much information to the manuscript. We have moved it to the appendix and replaced it with a new figure describing the spatial variation of near-surface temperatures within the landscape more explicitly as explained in a previous comment. We have also corrected the caption.

Fig. 5    I recommend explaining the variable abbreviations in the caption. That would help to make the figure more stand-alone, so readers don't have to look them up in the text. Also, I suggest renaming the response variable "surface T." to "near-surface T." to increase coherence with the text.

> We have clarified the caption and modified the figure as suggested.

Fig. 6    It is not clear from the plot if all vertical axes show beta or temperature differences. I suggest adding a label for clarification.

> All vertical axes do indeed represent beta but this should be clearer from the figure. We have clarified the axis titles.

I hope that the authors will find my remarks helpful. I wish them good luck and all the best!

Jonathan von Oppen

> Thank you for your thorough comments!

**Reviewer 2:**

The manuscript presents an analysis of winter near-surface temperatures along a south-north transect, going from boreal forest to tundra regions in Finland. The manuscript studies the drivers affecting the near-surface temperature in winter using a Structural Equation Modelling framework for the boreal and the tundra regions. Results show that snow cover duration has a strong control on soil temperature, but with opposite effect for mid-winter compared with late-winter. Site with shallow snowpack show stronger spatial variability, while site with flat topography and deep snow show strong decoupling between air temperature and soil temperature.

The manuscript is overall well written and generally show expected results. The findings of the study are not necessarily new; however, the dataset is quite extensive along a south-north gradient, which add some value to the study. The analysis is based on an interesting statistical approach which allows to see the interactions between the different drivers. However, such approach limits the applicability of the findings and in that sense a discussion on the possibility to use the dataset to improve physical modelling of snow and soil temperature could be interesting.

> Thank you for the feedback! We agree that combining this dataset with physical snow modelling would be interesting and is certainly something we have thought to research in the future. While we do not think the findings of this study directly benefit modelling in a concrete way, the possibilities of the dataset are now discussed briefly (L384-387).

Some aspects of the methodology should also be clarifier. Overall, the manuscript is suitable for The Cryopshere, but proper improvement should be brought to the manuscript.

1    The abstract can be clarified. For example, mentioning "seven study areas across boreal and tundra landscapes", it seems that the study was made across the northern hemisphere. It is important to clarify the study extent. There should be one or two sentence on the method used to get to the results (statistical approach).- Also, the results based on snow cover duration and the SEM is not quite clear in the abstract.

> Thank you for the suggestions on improving the abstract. We have clarified that the study domain covers Finland (L6). We have added that the results are based on empirical methods (L5-7) and clarified the results related to snow cover duration and SEMs (L11-14).

2    Line 38: "slow down snow melt during spring through energy balance controls". It is more complex than that. There are melting related to tree radiation around the trunk. It is mentioned in the discussion. Need to be clarify here.

This is a good point. The effect of canopy on below-canopy microclimates and thus snow melt is indeed more complex and we have expanded this (L41-44).

3    Figure 1. There is a need to clarify what represent c and d. Is it a histogram of all study sites for each study area?

The panels c and d show density curves of near-surface air temperatures (c) and snow cover duration (d) in all study sites within each study area. We have clarified this in the figure caption.

4    It is important in the text to well distinguish between "study area" and "study site". Sometimes, it can get confusion. Maybe using clear acronym for each could help?

Thank you for pointing this out. We have gone through the text carefully to make sure that it is always clear when we are discussing about study areas and when about study sites and what the discussed level of variation is in each sentence. However, we do not think that an acronym for each would make the text easier to understand but rather complicate it further.

5    Line 91: There is a need to clarify what "-6" means. Is it 6 centimeters under the surface. So it means that the 2 cm is above the surface? So it means that the means surface is 2 cm above the ground? Needs to be clarify. How the sensors were kept above the surface?

The sensors measure temperatures 6 cm below the ground, as well as 2 cm and 15 cm above the ground surface. When discussing near-surface temperatures, we mean near-surface air temperatures. We have clarified this focus on air rather than soil temperatures in the beginning of the manuscript (L1) and explained in more detail what the measured temperatures are (L104-105). The sensor is a stick-like logger which is carefully pushed to the soil to the correct depth and stays in place by itself (Wild et al. 2019).

6    Snow Cover duration: I have some doubt about the snow cover duration calculation. Why using the 15 cm? Even if the 15 cm is not cover by snow, it doesn't mean the 2 cm is not cover by snow? But the problem with using 2 cm to get the snow cover duration is that you would use the same measurements to get the snow cover duration and the impact of snow on near surface temperature. This point needs to be clarified/discussed.

This is a good point. It is true that there is some insulating effect from a thin snowpack as well. However, as the insulating capacity of a snow pack increases with increasing depth (e.g. Zhang 2005) and shallow snowpacks have been shown to poorly explain the temperature decoupling between temperatures below the snowpack and above it (Grundstein 2005), we decided to focus on periods with over 15 cm of snow as we expected that to better describe the buffering effect of a snowpack. We have clarified and further discussed this in L133-138.

7    In addition it is not clear if the snow cover duration was calculated for each "study site" or each "study area". Figure A1 is confusion because it shows all the near surface temperature at 2 cm (? need to be clarify) and the snow cover duration. However, that would be interesting to show the 15 cm temperature and air temperature to see how the snow cover duration was calculated.

The snow cover duration was calculated for each study site which has now been clarified (L132). In Figure A1, the bottom (top) line of the dark grey ribbon shows the daily minimum (maximum) near-surface temperatures measured at 15 cm height in the sites mentioned. We have clarified this in the caption. Air temperature was not used in the algorithm, so we do not think that adding it to the figure would help in understanding how the algorithm functions.

8    Line 147-150: These sentences are confusing. It seems that the calculations were done for each "study area". However, all the data is available to make the calculation at each "study

sites". From these sentences, I understand that the snow cover duration is calculated for a study area, when you can calculate it at each study site. It would be very important to clarify this point and clarify how many "N" are used in the SEM.

We understand that the sentence was confusingly written. The temperature variables as well as the snow cover duration were indeed calculated for each study site separately even though this was not explicitly mentioned in the text. In the calculation of the temperature variables, the timing of the "mid-winter" and "late winter" was defined for each study area collectively to keep this aspect of variation similar within the study areas. We have restructured this part to be more accurate and easier to understand (L186-193).

9  It also seems that the total snow cover duration is used as a variables in the SEM to explain the near-surface temperature in mid-winter. It seems inadequate to use a full winter snow cover duration as a variable to explain near-surface temperature in the middle of the winter? Maybe looking at the beginning of the snow cover would make more sense?

This is a good question and we understand that this point needs further explanation. As can be seen for example in figures 3 and A2, the beginning of the snow cover season is more uniform within the study areas and does not necessarily describe the accumulation patterns within the landscape as well as the total snow cover duration or snow melting date. Our reasoning for using the total snow cover duration is that this is likely to more accurately reflect where snow does and does not accumulate in a landscape, and therefore better represents the buffering effect of snow, as the more snow accumulates in a certain place, the longer it will take for it to melt as well. As can be seen in figure 5, the total snow cover duration does indeed correlate strongly with mid-winter near-surface air temperatures. However, we understand that this effect is not always obvious and regarding the melting, the whole picture is more complex, as noted in a previous comment concerning the effect of forest structure. To check our assumptions, we ran the mid-winter SEMs again with snow arrival date and have added the relevant results in the appendix (Table A2 and L193-195). We have also further discussed this in L349-353.

10  Would be important to mentioned if the study area are in permafrost regions. It will have a impact on the thermal regime of the soil and thus on the near-surface temperature.

This is a good point as the most northern areas of the study domain are close to permafrost regions. There is no permafrost within the actual study areas and we have clarified this (L90).

11  Would be useful to give a more representative acronym for the "beta".

We have the acronym to slope which should also describe the variable appropriately.

12  line 207: "Snow cover duration had a strong positive effect (0.73) in mid-winter". It is quite surprising to get such a strong relation when the end of snow season should not have any impact on the mid-winter soil temperature?

As we explained above, while it is true that the end of the snow cover season does not directly affect mid-winter temperatures, it does reflect snow accumulation patterns which do strongly control mid-winter conditions. As mentioned in a previous comment, we have clarified this in (L349-353).

13  Figure 6: Should clarify what is on Y axis. Also "linear regression model calculated from a two-week moving window", add "(beta)"

The y-axis in all the panels shows the slope. We have clarified this and modified the caption as suggested.

14  In the discussion, it will be important to mention the soil thermal regime. Your measurement are above the ground (2 cm) if I understood well. However, it is well known that a wet soil will stay at zero curtain longer because of the latent heat. Permafrost can also alter the soil thermal regime. Even if the measurement are not done in the soil, the author have to recognize the potential strong impact of soil on the results, which are not considered in the study.

This is an excellent point. While considering soil-related processes governing near-surface temperatures was outside the scope of this study, the soil thermal regime and its variation is important and we look

forward to addressing it in more detail in a future study. Here we have, as suggested, added discussion on the impact of soil properties, including soil moisture, on near-surface air temperatures (L328-332).

15  As mentioned earlier, the results are interesting, but not quite new. Ideally, the study would have been conducted using snow physical modelling. But I understand that it is not the scope of the work. However, should be important to relate the results to possible improvement in soil temperature modeling.

While the approach of this manuscript was empirical, we agree that it would be interesting to combine our dataset with a physical snow model. In our opinion, the most relevant finding of this manuscript, concerning snow and winter microclimate modelling, is the considerable magnitude of landscape-level variation and the need to take this variation into account in order to accurately simulate snow cover and its impact on winter microclimates (L385-387). More concrete improvements of snow modelling would, in our opinion, require a different approach than the one in this manuscript but we look forward to addressing this in future studies.

16. Figure A2: not clear what "predicted" mean in that context?

Predicted here refers to the prediction made by the snow cover algorithm. We have clarified this in the caption.

Thank you for your thorough comments!